# Extending Graph Condensation to Multi-Label Datasets: A Benchmark Study

**Liangliang Zhang**                                                                  *zhangl41@rpi.edu*
*Rensselaer Polytechnic Institute*

**Haoran Bao**                                                                         *baoh2@rpi.edu*
*Rensselaer Polytechnic Institute*

**Yao Ma**                                                                            *may13@rpi.edu*
*Rensselaer Polytechnic Institute*

**Reviewed on OpenReview:**

## Abstract

As graph data grows increasingly complicated, training graph neural networks (GNNs) on large-scale datasets presents significant challenges, including computational resource constraints, data redundancy, and transmission inefficiencies. While existing graph condensation techniques have shown promise in addressing these issues, they are predominantly designed for single-label datasets, where each node is associated with a single class label. However, many real-world applications, such as social network analysis and bioinformatics, involve multi-label graph datasets, where one node can have various related labels. To deal with this problem, we extend traditional graph condensation approaches to accommodate multi-label datasets by introducing modifications to synthetic dataset initialization and condensing optimization. Through experiments on eight real-world multi-label graph datasets, we prove the effectiveness of our method. In the experiment, the GCond framework, combined with K-Center initialization and binary cross-entropy loss (BCELoss), generally achieves the best performance. This benchmark for multi-label graph condensation not only enhances the scalability and efficiency of GNNs for multi-label graph data but also offers substantial benefits for diverse real-world applications. Code is available at https://github.com/liangliang6v6/Multi-GC.

## 1 Introduction

Graph-structured data are fundamental to many real-world applications, including social networks, academic citation networks, chemical molecules, protein-protein interaction networks, mapping services, and product recommendation systems (Battaglia et al., 2018; Wu et al., 2020; Zhou et al., 2020; Wu et al., 2022). In these graph structures, nodes represent entities (e.g., users in social networks), while edges represent relationships between these entities (e.g., social connections between users)(Zhang et al., 2023).

Graph Neural Networks (GNNs) are a class of deep learning models specifically designed to process and learn from graph-structured data. It utilizes the topology of the graph to capture dependencies and learn meaningful nodes, edges, and graph-level representations. By extending deep learning to non-Euclidean domains, GNNs have achieved state-of-the-arts performance in many graph machine learning applications (Zhang et al., 2023; Hamilton et al., 2018). For example, in the recommendation system (Li et al., 2019; Mao et al., 2021; Wu et al., 2022; Zhang et al., 2019a), drug discoveries (Jiang et al., 2021; Zhang et al., 2021), fraud account detection (Dou et al., 2020; Wang et al., 2021) and traffic forecasting (Jiang & Luo, 2022). GNNs combine the node features and structure of graph as the input and output the representations of graph information (Hamilton et al., 2018; Velickovic et al., 2017). Among all the GNNs techniques, the most broadly followed methods is recursive neighborhood aggregation scheme, which aims to get the representation

of each node, respectively (Xu et al., 2018). By applying different neighborhood aggregation and graph-level pooling methods, GNNs are capable of handling various of tasks like node classification (Xiao et al., 2022), link prediction (Hasan & Zaki, 2011), and graph classification (Zhang et al., 2018).

Even though GNNs(Zhang et al., 2023; 2019b) are widely used for analyzing graphs, training these models on large-scale graphs is still computationally expensive (Gao et al., 2018; Bojchevski et al., 2020). Furthermore, nowadays, graph data is becoming gigantic with numerous nodes and edges in a single graph (Zhang et al., 2023). For example, the Twitter user graph has 288M monthly active users as of 3/2015 and an estimated average of 208 followers per user for an estimated total of 60B followers (edges) (Ching et al., 2015). Such industry graphs can be two orders of magnitude larger, like hundreds of billions or up to one trillion edges. In other words, the complexity and large scale of graph data pose significant challenges for training GNNs(Jin et al., 2022b). Despite the scalability challenges, for some specific applications that need re-training multiple times models would encounter more hardships rather than computation cost.

To address these large-scale datasets challenges, graph condensation methods recently (Xu et al., 2024; Gao et al., 2024) have gained lots of attention. Graph condensation is motivated by condensing a large graph dataset into a smaller informative synthetic graph, thereby enhancing the scalability and efficiency of GNNs (Xu et al., 2024; Gao et al., 2024; Hashemi et al., 2024). The goal is to achieve comparable performance in GNNs using this synthetic small graph, instead of relying on the original large-scale graph (Loukas, 2018; Jin et al., 2022a). By doing so, condensed synthetic graph enables GNNs to maintain strong predictive performance while significantly reducing the computational resources required for tasks such as training and inference. By eliminating redundant information in original dataset, it makes the synthetic graph more manageable within the constraints of limited computation resources, thereby providing better support for graph data mining tasks and applications such as Continual learning (Liu et al., 2023), Network Architecture Search (NAS)(Gao et al., 2021), etc. Moreover, take node classification task as an example, the node can be well classified is because GNNs have learned to capture the unique pattern of nodes to distinguish them from other nodes in different classes.

Existing graph condensation methods (Xu et al., 2024; Gao et al., 2024) explore diverse strategies such as gradient distance matching (Jin et al., 2022b; Yang et al., 2023), trajectory matching (Zheng et al., 2024; Zhang et al., 2024), kernel ridge regression (Xu et al., 2023; Wang et al., 2024), and distribution matching (Liu et al., 2022; 2023). However, in many real-world applications, nodes are often associated with multiple labels (Huang & Zhou, 2012; Read et al., 2011). For example, users in social networks may belong to multiple groups reflecting diverse interests, articles in citation networks can span multiple research topics, and proteins in biological networks may participate in various processes, such as molecular activities or cellular functions (Shi et al., 2019; Akujuobi et al., 2019; Zeng et al., 2019). Nodes in multi-label graphs not only possess features, but are also associated with multiple overlapping class labels (Shi et al., 2019; 2020; Akujuobi et al., 2019; Zhao et al., 2023). Existing graph condensation methods are designed for single-label settings. Directly applying single-label methods to multi-label datasets leads to the loss of critical label interaction and distribution information, as the underlying class-wise for single label assumption does not hold.

To bridge this gap from single-label to multi-label setting, in this work, we adapt available graph condensation techniques to the mainstream multi-label datasets. Our approach includes modifications to the synthetic dataset initialization and original datasets condensation matching stages. By evaluating various adaptation settings, we identify three key insights into the optimal combination of initialization and optimization methods. Using these best settings, we further analyze graph condensation models and present three additional observations, highlighting the unique challenges of condensing multi-label graph datasets. Our contributions are as follows:

1. We extend classic SOTA graph condensation methods, including GCond, GCDM, and SGDD, to the multi-label graph dataset scenario by introducing multi-label synthetic dataset initialization and condensing optimization methods.

2. To find the best adaption strategies, we compare different initialization techniques such as Random sampling, Herding, K-Center and probability synthetic multi-label combine with two different multi-label loss functions – SoftMarginLoss and BCELoss.

3. With the best adaption settings, we evaluate the condensation methods with F1-micro and F1-macro scores on eight real-world multi-label datasets: PPI, PPI-large, Yelp, DBLP, PCG, HumanGo, EukaryoteGo, and OGBN-Proteins. Finally, we find the GCond method generally works best with K-Center initialization and BCELoss.

## 2 Related Work

### 2.1 Graph Neural Networks

Graph Neural Networks (GNNs) are a class of deep learning models specifically designed to process and learn from graph-structured data. GNNs have achieved outstanding performance in many applications (Zhang et al., 2023; Hamilton et al., 2018). For instance, in the recommendation system on social network (Li et al., 2019; Mao et al., 2021; Wu et al., 2022; Zhang et al., 2019a), drug discoveries from molecule graphs (Jiang et al., 2021; Duvenaud et al., 2015; Zhang et al., 2021), fraud account detection on financial graphs (Dou et al., 2020; Wang et al., 2021) and traffic forecasting in transportation graph (Jiang & Luo, 2022).

GNNs combine the node features and structure of graph as the input and output the representations of graph (Hamilton et al., 2018; Velickovic et al., 2017; Kipf & Welling, 2016). Among all the GNNs techniques, they broadly follow a recursive neighborhood aggregation scheme to get the representation of each node (Xu et al., 2018). Some of the typical GNNs models like the Graph Convolutional Network (GCN) (Kipf & Welling, 2016) operate by aggregating and transforming information from neighboring nodes layer by layer. Variants like Graph Attenstion Nesworks (GAT) (Velickovic et al., 2017) use attention mechanisms to prioritize certain neighbors, while GraphSAGE (Hamilton et al., 2017) samples neighborhoods to scale to large graphs. By applying different neighborhood aggregation and graph-level pooling methods, GNNs are capable of handling various of tasks (Xiao et al., 2022; Hasan & Zaki, 2011; Zhang et al., 2018) like node classification, link prediction, and graph classification.

### 2.2 Multi-Label Classification on Graphs

Multi-label classification on graphs (Zhao et al., 2023) is a fundamental task in graph machine learning, where nodes in a graph are associated with multiple labels simultaneously. This task is critical in numerous real-world applications, including social networks, bioinformatics, and recommendation systems. For instance, in social networks, users may belong to multiple interest groups, while in bioinformatics, proteins may participate in multiple biological processes such as molecular functions and cellular activities (Huang & Zhou, 2012; Read et al., 2011; Shi et al., 2019; Akujuobi et al., 2019). In multi-label classification, the goal is to predict a set of labels for each data point in the dataset. The labels can be thought of as binary variables, where a label is either present (1) or absent (0). Recently, there have been several approaches that use neural networks for multi-label classification.

Traditional multi-label classification methods relied on adapting single-label classification techniques to multi-label scenarios. Popular strategies include problem transformation methods (e.g., binary relevance and classifier chains) and algorithm adaptation approaches, which extend existing classifiers to handle multiple labels directly (Read et al., 2011; Tsoumakas & Katakis, 2008). These approaches use architectures such as recurrent neural networks (RNNs) and convolutional neural networks (CNNs) to learn representations of the input data and make predictions for the labels. However, these approaches were not designed to exploit the graph structure and are therefore suboptimal for graph data.

With the advent of GNNs, researchers began exploring multi-label classification directly on graph-structured data. GNNs excel in capturing the relational and topological information of graphs, making them well-suited for multi-label classification tasks. For example, GraphSAGE (Hamilton et al., 2017) and GAT (Velickovic et al., 2017) utilize neighborhood aggregation to generate node embeddings, which can be fed into multi-label classifiers. Specialized models like ML-GCN (Chen et al., 2019) integrate graph convolutional layers with multi-label prediction heads to improve performance on multi-label datasets.

### 2.3 Graph Condensation

Despite the impressive performance of GNN models, training them on large-scale datasets is often challenging due to high computational costs, especially when retraining or dealing with redundant information. To address these issues, graph condensation techniques have been developed. These methods distill a large-scale graph into a smaller yet informative synthetic graph, improving the scalability and efficiency of GNNs (Xu et al., 2024; Gao et al., 2024; Hashemi et al., 2024). The condensation process generates a compact representation that preserves essential topological and feature-based properties from the original graph (Loukas, 2018; Jin et al., 2022a). This allows GNNs to achieve comparable performance using the condensed graph while significantly reducing the computational resources needed for training and inference.

Existing graph condensation works (Xu et al., 2024; Gao et al., 2024) investigate different condensing strategies including gradient distance matching(Jin et al., 2022b; Yang et al., 2023), trajectory matching(Zheng et al., 2024; Zhang et al., 2024), kernel ridge regression(Xu et al., 2023; Wang et al., 2024), and distribution matching(Liu et al., 2022; 2023). Condensed graphs facilitate more efficient model execution without sacrificing the quality of learning, making this technique a promising method for real-world GNN deployments. By eliminating redundant information in original dataset, graph condensation makes the synthetic graph more manageable within the constraints of limited computation resources, thereby providing better support for graph data mining tasks and applications such as Continual learning (Liu et al., 2023) and Network Architecture Search (NAS)(Gao et al., 2021), etc. Moreover, take node classification as an example, the reason a node can be well classified is that GNNs have learned to capture the unique pattern of nodes to distinguish them from other nodes in different classes.

## 3 Problem Formulation

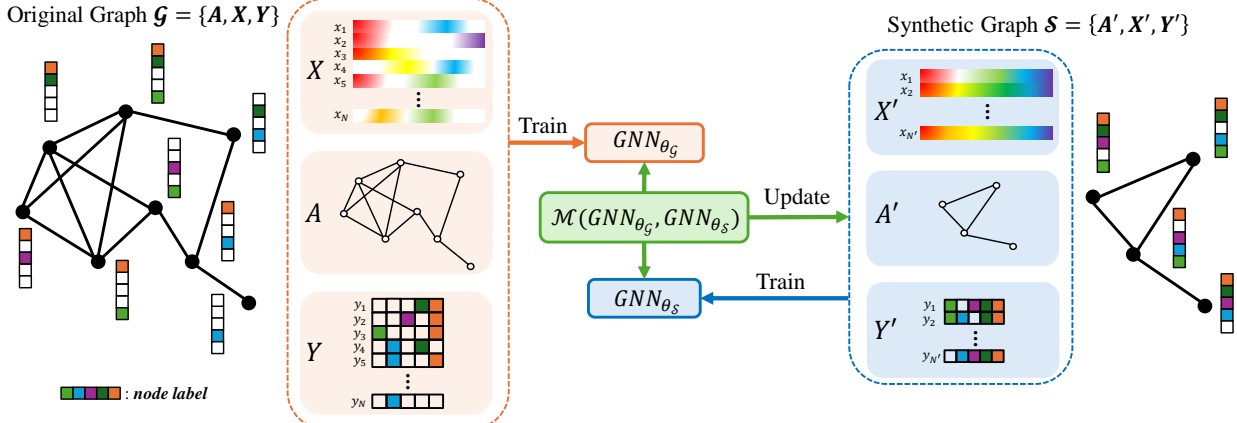

Figure 1: Workflow for multi-label graph condensation. It shows the process of condensing a large multi-label graph $\mathcal{G} = \{A, X, Y\}$ into a smaller synthetic graph $\mathcal{S} = \{A', X', Y'\}$, where $Y$ and $Y'$ represent the multi-label matrix. Various matching strategies, denoted by $\mathcal{M}(\cdot)$, are employed to ensure that key information in the original graph is captured. Our goal is to use the synthetic graph $\mathcal{S}$ to train a GNN that achieves comparable performance to the one trained on the original graph $\mathcal{G}$, thus reducing graph size while retaining performance.

Consider a graph $\mathcal{G} = \{A, X, Y\}$, where $N$ is the number of nodes, $A \in \mathbb{R}^{N \times N}$ is the corresponding adjacency matrix, $X = \{x_1, x_2, \cdots, x_N\} \in \mathbb{R}^{N \times d}$ is the $d$-dimensional node feature matrix. In multi-label scenario, each node is associated with multiple labels revealing characteristics or semantics of the node. In this case, the label matrix can be represented as $Y \in \{y_1, y_2, \cdots, y_N\} \in \{0, 1\}^{N \times K}$, where $K$ is the number of classes. Following the definition of graph condensation, our goal is to learn a smaller synthetic graph denoted as $\mathcal{S}$, which can contain the ability to train downstream tasks' GNNs and achieve competitive performance compared with the original large-scale complex graph $\mathcal{G}$. Similarly, the synthetic graph can be denoted

as $\mathcal{S} = \{A', X', Y'\}$ along with $A' \in \mathbb{R}^{N' \times N'}$, and $X' \in \mathbb{R}^{N' \times d}$, $Y' \in \{0, 1\}^{N' \times K}$. During this stage, our expectation is that the scale of the synthetic graph can be much smaller compared with the original graph, denotes as $N' \ll N$.

In this work, we adapt the graph condensation techniques to a multi-label setting, trying to get the synthetic multi-label graph. First, we introduce the general framework of graph condensation in Figure 1. Generally, graph condensation methods are using different matching strategies to optimize the synthetic graph $\mathcal{S}$ similar to the original graph $\mathcal{G}$. Despite these various strategies, the key point is to keep the comparable performance of GNNs while training on small graph. Therefore, our objective can be formulated as follows (Jin et al., 2022b):

$$\mathcal{S}^* = \arg\min_{\mathcal{S}} \mathcal{M}(\theta_{\mathcal{S}}^*, \theta_{\mathcal{G}}^*) \quad \text{s.t.} \quad \theta^* = \arg\min_{\theta} \mathcal{L}(GNN_\theta), \tag{1}$$

where $GNN_\theta$ denotes a GNN parameterized with $\theta$. Specifically, $\theta_{\mathcal{S}}$ and $\theta_{\mathcal{G}}$ are the parameters trained on graph $\mathcal{S}$ and $\mathcal{G}$, respectively. $\mathcal{M}(\cdot)$ is the matching strategy used to match the model $\theta_{\mathcal{S}}^*$ trained synthetic graph $\mathcal{S}$ to the one $\theta_{\mathcal{G}}^*$ trained on the original graph $\mathcal{G}$. $\mathcal{L}(\cdot)$ is the loss function used to measure the difference between model predictions and ground truth. For most graph condensation methods, the loss function is cross-entropy loss.

As the specific initialization of model parameters $\theta$ can lead to the overfitting problem (Jin et al., 2022b), one common solution (Wang et al., 2018) is to generate the synthetic data following a distribution of random initializations of $P_{\theta_0}$. Furthermore, the reformulated optimization problem as follows:

$$\mathcal{S}^* = \arg\min_{\mathcal{S}} \mathbb{E}_{\theta_0 \sim P_{\theta_0}}[\mathcal{M}(\theta_{\mathcal{S}}^*, \theta_{\mathcal{G}}^*)$$
$$\text{s.t.} \quad \theta^* = \arg\min_{\theta(\theta_0)} \mathcal{L}(GNN_\theta)]. \tag{2}$$

where $\theta(\theta_0)$ indicates $\theta$ is a function acting on $\theta_0$.

## 4 Graph Condensation Methods

Building on the global understanding of existing single-label graph condensation methods, we observe that while different graph condensation methods differ primarily in their matching strategies, the overall workflow remains largely consistent. Therefore, before adapting these methods to the multi-label scenario, we will illustrate the different matching strategies of condensation methods first.

### 4.1 GCond: Gradient Matching

By computing the gradient of $\mathcal{L}$ w.r.t. $\mathcal{S}$ and optimize $\mathcal{S}$ via gradient descent. For each training step, the $opt_\theta$ is the update rule, consider the one-step gradient descent for model parameters:

$$\theta_{t+1}^{\mathcal{S}} \leftarrow \theta_t^{\mathcal{S}} - \eta \nabla_\theta \mathcal{L}(GNN_{\theta_t^{\mathcal{S}}}(A', X'), Y')$$
$$\theta_{t+1}^{\mathcal{G}} \leftarrow \theta_t^{\mathcal{G}} - \eta \nabla_\theta \mathcal{L}(GNN_{\theta_t^{\mathcal{G}}}(A, X), Y) \tag{3}$$

where $\theta_t^{\mathcal{S}}$ and $\theta_t^{\mathcal{G}}$ denote the model parameters trained on $\mathcal{S}$ and $\mathcal{G}$ at step $t$. $\eta$ is the learning rate for the gradient descent.

By matching the training gradient trajectory, define distance function $D(\cdot, \cdot)$, define the gradient matching strategy $\mathcal{M}_{GCond}$ as:

$$\mathcal{M}_{GCond} = D(\nabla_\theta \mathcal{L}(GNN_{\theta_t^{\mathcal{S}}}(A', X'), Y'),$$
$$\nabla_\theta \mathcal{L}(GNN_{\theta_t^{\mathcal{G}}}(A, X), Y)) \tag{4}$$

Assume $T$ is the number of steps of the whole training, the final optimized objective is:

$$\mathcal{S}^* = \arg\min_{\mathcal{S}} \mathbb{E}_{\theta_0 \sim P_{\theta_0}}[\sum_{t=0}^{T-1} D(\nabla_\theta \mathcal{L}(GNN_{\theta_t^{\mathcal{S}}}(A', X'), Y'),$$
$$\nabla_\theta \mathcal{L}(GNN_{\theta_t^{\mathcal{G}}}(A, X), Y))]. \tag{5}$$

Direct joint learning can be highly challenging because the three variables $A', X'$, and $Y'$ in synthetic graph are interdependent. One straightforward way is to treat $A'$ and $X'$ as free parameters. By modeling the synthetic graph structure as a function of the condensed node features, synthetic graph structure $A'$ can be learned by:

$$A' = g_\phi(X') \tag{6}$$

where $\phi$ is the parameter of a multi-layer neural network. So the rewrite objective is:

$$\mathcal{S}^* = \arg\min_{X',\phi} \mathbb{E}_{\theta_0 \sim P_{\theta_0}} \Big[ \sum_{t=0}^{T-1} D(\nabla_\theta \mathcal{L}(GNN_{\theta_t^\mathcal{S}}(g_\phi(X'), X'), Y'),$$
$$\nabla_\theta \mathcal{L}(GNN_{\theta_t^\mathcal{G}}(A, X), Y)) \Big] \tag{7}$$

## 4.2 SGDD: Structure-broadcasting

Instead of treating $A'$ and $X'$ in the synthetic graph $\mathcal{S}$ as free parameters, Yang et al. (2023) points out that this way ignores the original structure $A$ in $\mathcal{G}$. To address this problem, they broadcast $A$ as supervision for the generation of $A'$. By introducing graph on (Gao & Caines, 2019; Ruiz et al., 2020; Xia et al., 2023) to matching the different shapes between $A$ and $A'$, SGDD use random noise $\mathcal{Z}(N') \in \mathbb{R}^{N' \times N'}$ as input coordinates. Through the generative model to synthesize an adjacency matrix $A'$ with $N'$ nodes, the process can be formulated as:

$$A' = GEN(\mathcal{Z}(N'); \Phi) \tag{8}$$

where $GEN(\cdot)$ is the generative model with parameter $\Phi$. Then the structure optimization is:

$$\mathcal{M}_{structure} = Dis(A, GEN(\mathcal{Z}(N'); \Phi)), \tag{9}$$

where $A$ is supervision and $Dis(\cdot)$ is a metric that measure the difference between $A$ and $A'$. By adding the corresponding nodes information from $X'$ and $Y'$ to the inherent relation learning (Pfeiffer III et al., 2014; Shalizi & Thomas, 2011), the revised generation model is $GEN(\mathcal{Z}(N') \oplus X' \oplus Y'; \Phi)$, $\oplus$ denotes the concatenate operation. In general, SGDD concurrently optimizes the parameters $X'$ and $A'$. While the refinement of $X'$ is achieved through a gradient matching strategy, whereas the $A'$ is enhanced using the Laplacian energy distribution(LED) matching technique (Tang et al., 2022; Das et al., 2016; Gutman & Zhou, 2006). During each step, the other component is frozen to ensure effective refinement, and the overall training loss function can be summarized as:

$$\mathcal{M}_{SGDD} = \mathcal{M}_{GCond} + \alpha \mathcal{M}_{structure} + \beta ||A||_2, \tag{10}$$

where $||A||_2$ is proposed as a sparsity regularization term, $\alpha$ and $\beta$ are trade-off parameters.

## 4.3 GCDM: Distribution Matching

Inspired by the distribution matching strategy in image dataset condensation (Zhao & Bilen, 2023), (Liu et al., 2022) adapt the receptive field distribution matching method in GNN. In the graph dataset, with the reproducinf kernal Hilbert space $\mathcal{H}$ defined by GNNs and with in each class $k$, GCDM optmize the maximum mean discrepancy (MMD). Let $\Phi_\theta$ be an $L-$layer GNN model parameterized by $\theta$, the loss of GCDM is:

$$\mathcal{M}_{GCDM} = \min_{\phi, X'} \sum_{k=0}^{K-1} r \cdot \max_{\theta_c} \left\| \frac{1}{V_k} \sum_{i \in V_k} \text{emb}_i^k - \frac{1}{V_k'} \sum_{j \in V_k'} \text{emb}_j^k \right\|_2^2, \tag{11}$$

where the node embeddings are:

$$\{\text{emb}_i^k\}_{i=0}^{N-1} \leftarrow \Phi_{\theta_c}(A, X) \tag{12}$$

$$\{\text{emb}_j^k\}_{j=0}^{N'-1} \leftarrow \Phi_{\theta_c}(A' = g_\phi(X'), X') \tag{13}$$

$V_k$ and $V_k'$ represent the node sets belong to the class $k$ of original graph $\mathcal{G}$ and synthetic graph $\mathcal{S}$, respectively. $K$ is the number of multi-label classes. $r$ denotes the condensed ratio of the synthetic graph.

## 5 Datasets and Experiment Settings

In this section, we first present the multi-label graph datasets used in our benchmark investigation, accompanied by the detailed introduction of label information in real-world application. Next, we detail the evaluation metrics used to assess the effectiveness of the adaptation methods, providing a robust framework for performance comparison.

**Datasets.** Specifically, we employ eight real-world datasets including PPI (Zeng et al., 2019) , PPI-large (Zeng et al., 2019) , Yelp(Zeng et al., 2019), DBLP (Akujuobi et al., 2019), OGBN-Proteins (Hu et al., 2020), PCG (Zhao et al., 2023), HumanGo (Chou & Shen, 2007; Liberzon et al., 2015), EukaryoteGo (Chou & Shen, 2007; Liberzon et al., 2015). The details of those datasets are introduced in Appendix A.

We follow the predefined data splits from (Zeng et al., 2019; Zhao & Bilen, 2023; Hu et al., 2020; Chou & Shen, 2007). 1 presents an overview of the datasets' characteristics.

Table 1: Multi-Label Graph Dataset Statistics

| Dataset | #Nodes | #Edges | #Avg.Edges | #Features | #Labels | Train/Val/Test |
|---|---|---|---|---|---|---|
| **PPI** | 14,755 | 225,270 | 15.27 | 50 | 121 | 0.66/0.12/0.22 |
| **PPI-large** | 56,944 | 818,716 | 14.38 | 50 | 121 | 0.79/0.11/0.10 |
| **Yelp** | 716,847 | 6,977,410 | 9.73 | 300 | 100 | 0.75/0.10/0.15 |
| **DBLP** | 28,702 | 68,335 | 2.38 | 300 | 4 | 0.60/0.20/0.20 |
| **OGBN-Proteins** | 132,000 | 39,000,000 | 295.45 | 8 | 112 | 0.66/0.16/0.18 |
| **PCG** | 3,233 | 37,351 | 11.55 | 32 | 15 | 0.60/0.20/0.20 |
| **HumanGo** | 3,106 | 18,496 | 5.96 | 32 | 14 | 0.60/0.00/0.40 |
| **EukaryoteGo** | 7,766 | 13,818 | 1.78 | 32 | 22 | 0.60/0.00/0.40 |

After introduce the single-label graph condensation methods, we will focus on adjusting key components to fit the multi-label task while preserving the core matching mechanisms. This ensures that our framework can be extended to other existing graph condensation techniques, allowing for flexible and efficient adaptation across different methods.

This benchmark presents a thorough evaluation of the various settings and methods used for graph condensation in the multi-label scenario. To evaluate the performance of adapted graph condensation methods, we utilize two widely recognized metrics in multi-label classification: F1-micro and F1-macro (Lipton et al., 2014). The calculation details are listed in Appendix A. We structure the benchmark results into three key parts: (1) identifying the best initialization and loss function settings, (2) comparing the full results of all graph condensation methods across multiple datasets, and (3) analyzing the results through multi-label class distribution and label correlation.

## 6 Adapt Existing Graph Condensation Methods for Multi-label Datasets

After the introduction of different graph condensation methods based on the matching strategies in Section 4 including GCond, SGDD and GCDM, we apply the above methods in multi-label graph datasets in this section. However, these methods could not directly be applied to condense multi-label graphs majorly due to two reasons: (1) The synthetic graph initialization is designed for single-class augmentation with pre-defined condensed labels. (2) The condensation methods depend on GNNs to match the original graph $\mathcal{G}$ and synthetic graph $\mathcal{S}$, the classification objectives are designed for single-label node classification. Therefore, in this section, we would like to adapt the existing graph condensation methods by investigating different adaptation strategies to address these two issues. In particular, we first describe the existing issues and the corresponding resolvents in Section 6.1. Then, we conduct experiments to investigate which strategies work best for the multi-label graph condensation tasks in Section 6.2.

| Datasets | C-rate | Random | | Herding | | K-Center | | Probability | Whole Dataset |
|---|---|---|---|---|---|---|---|---|---|
| | | Subgraph | Nodes | Subgraph | Nodes | Subgraph | Nodes | | |
| **PPI** | **1.00%** | 47.05 | **40.49** | 40.93 | 39.47 | **47.77** | 36.60 | 40.04 | 51.26 |
| **Yelp** | **0.02%** | 34.21 | 30.35 | 32.77 | 30.96 | **35.60** | **31.46** | 20.95 | 37.97 |
| **DBLP** | **0.80%** | **63.56** | **56.03** | 41.54 | 41.02 | 60.36 | 55.62 | 41.35 | 87.55 |
| **OGBN-Proteins** | **0.1%** | 14.12 | 15.44 | 11.62 | 15.36 | **29.95** | **24.81** | 15.53 | 18.86 |
| **PCG** | **4%** | 13.94 | 19.01 | **27.76** | 16.35 | 25.13 | **22.18** | 21.69 | 42.26 |

Table 2: F1-Micro Score (%) of Coreset Method with Different Initialization Strategies

| Datasets | C-rate | Random | | Herding | | K-Center | | Probability | | Whole Dataset |
|---|---|---|---|---|---|---|---|---|---|---|
| | | Without $A'$ | With $A'$ | Without $A'$ | With $A'$ | Without $A'$ | With $A'$ | Without $A'$ | With $A'$ | |
| **PPI** | **1.00%** | **48.55** | **50.17** | 46.71 | 47.80 | 35.99 | 48.38 | 44.62 | 43.31 | 51.26 |
| **Yelp** | **0.02%** | 30.16 | 29.53 | 30.48 | 32.23 | **34.50** | **34.90** | 21.33 | 22.10 | 37.97 |
| **DBLP** | **1%** | 52.10 | 57.48 | 45.95 | 47.12 | **52.56** | **60.77** | 48.24 | 48.56 | 87.55 |
| **OGBN-Proteins** | **0.10%** | 22.03 | 28.69 | 20.54 | 25.19 | **28.09** | **29.04** | 22.10 | 27.01 | 30.59 |
| **PCG** | **4%** | 20.50 | 23.79 | **22.14** | 25.38 | 19.88 | 22.36 | 22.07 | **27.21** | 42.26 |

Table 3: F1-Micro Score (%) of GCond Method with Random/Herding/K-Center/Probability Distribution Initialization with/without Learning from Structure for SoftMarginLoss.

## 6.1 Multi-label Adaptation

Most graph condensation methods, such as GCond, utilize batch sampling by class to optimize synthetic graphs. This involves initializing synthetic graph labels $Y'$ based on predefined class-wise condensation rates $r$. To simplify computation and reduce memory usage, the loss of gradient matching is typically calculated separately for nodes of different classes. Although this approach is effective for single-label graphs, where each node belongs to only one class, it encounters challenges in multi-label settings, where nodes are associated with multiple labels simultaneously.

One major challenge is in the classification of a single label graph, each node belongs to a specific class, which facilitates straightforward batch sampling in class. However, in multi-label graphs, nodes often have overlapping class memberships, complicating node partitioning for optimization while preserving the label's cooccurrence. Redundant node sampling worsens the problem by repeatedly including the same nodes in different subgraphs, which can bias certain label associations while ignoring others. It can undermine the integrity and representational accuracy of the synthetic graph. In multi-label graphs, the frequency of class in multi-label could be imbalance. Traditional sampling can worsen this, leading to poor representations for rare labels and distorting the original label distribution. Furthermore, single-label loss functions are not well-suited for multi-label scenarios. To ensure accurate gradient optimization and preserve the original multi-label structure, the loss function must effectively capture label dependencies.

To address these challenges, we explore various initialization and optimization methods to evaluate their impact on preserving multi-label relationships and classification accuracy. Our focus is on two key areas: **initialization** and **optimization** of synthetic graphs, ensuring the essential multi-label features of the original graph are effectively captured.

**Issue 1: Initialization of Synthetic Multi-Label Graph.** In single-label settings, the condensed label $Y'_{single}$ is predefined using ground-truth class distribution and retrieves $X'$ randomly from each class.

As most of the graph condensation methods adapt Eq.6 to learn synthetic graph structure $A'$ from node features $X'$, one straightforward method for multi-label adaptation is sampling the nodes from original

| Datasets | C-rate | Random | | Herding | | K-Center | | Probability | | Whole Dataset |
|---|---|---|---|---|---|---|---|---|---|---|
| | | Without $A'$ | With $A'$ | Without $A'$ | With $A'$ | Without $A'$ | With $A'$ | Without $A'$ | With $A'$ | |
| **PPI** | **1.00%** | **49.95** | **51.35** | 47.47 | 48.23 | 36.79 | 49.27 | 45.65 | 43.91 | 51.26 |
| **Yelp** | **0.02%** | 29.24 | 30.76 | 31.76 | 32.41 | **32.36** | **33.94** | 22.96 | 25.62 | 37.97 |
| **DBLP** | **1%** | 53.41 | 59.40 | 46.87 | 48.50 | **54.21** | **70.74** | 49.73 | 51.33 | 87.55 |
| **OGBN-Proteins** | **0.10%** | 24.65 | 28.88 | 21.45 | 26.21 | **28.15** | **29.40** | 21.73 | 27.11 | 30.59 |
| **PCG** | **4%** | 22.26 | 25.45 | **24.66** | 26.76 | 20.17 | 25.58 | 23.47 | **28.28** | 42.26 |

Table 4: F1-Micro Score (%) of GCond Method with Random/Herding/K-Center/Probability Initialization with/without Learning from Structure for BCELoss.

datasets. Inspired by the heuristic sampling algorithms of baseline Coreset methods used in GCond, the sampled subgraph by different strategies can be used for the synthetic multi-label initialization to get a better performance.

As some of the coreset methods gain comparable performance as baseline (Welling, 2009; Farahani & Hekmatfar, 2009; Sener & Savarese, 2017), we apply those heuristic algorithms to extract $N'$ nodes from the full set of nodes to obtain subgraphs $S'$ with corresponding condensation rates as the initialization methods.

- **Random selection** is sampling the nodes randomly, probability of selecting node $i$ is $P(i) = \frac{1}{N}$.

- **Herding** (Welling, 2009) is more structured compared with random selection strategy. It is based on iterative minimization of the distance between the mean of the selected nodes' feature vectors and the mean of the entire graph feature vectors. Let $\mu_{\mathcal{G}}$ present the mean feature vector of the original graph: $\mu_{\mathcal{G}} = \frac{1}{N} \sum_{i=0}^{N-1} x_i$ and $\mu_{\mathcal{S}} = \frac{1}{N'} \sum_{j=0}^{N'-1} x'_j$

- **K-Center** (Farahani & Hekmatfar, 2009; Sener & Savarese, 2017) aims to maximize the coverage of the original graph feature space by minimizing the maximum distance between unselected nodes and their closest selected node. Let $\mathcal{S} = \{s_1, s_2, \cdots, s_{N'}\}$ be the selected node set. The goal is to select nodes such that for every unselected node $i$, the distance to the nearest selected node $s_j \in \mathcal{S}$ is minimized: $\min_{s_1, \cdots, s_{N'}} \max_{i \in \{1, \cdots, N\}} \min_{j \in \{1, \cdots, N'\}} \|x_i - x_{s_j}\|_2$

- **Probabilistic synthetic** ensures the selected subgraph maintains the label distribution of the original graph. For each class $k$, we define the label distribution as the fraction of nodes in the graph that belongs to class $k$: $p_k = \frac{1}{N} \sum_{i=1}^{N} Y_{i,k}$

Furthermore, the structure $A'$ of synthetic graph is optimized from nodes features $X'$. Based on the researches (Ding et al., 2022; Zhu et al., 2021; Jin et al., 2022b), this "Graphless" learning process provides competitive results and facilitate various downstream tasks. To systematically evaluate the impact of this aspect, we clearly differentiate between "node-only" initialization methods (which do not consider structural information) and "subgraph-based" methods (which explicitly incorporate graph structure). By comparing these two categories throughout our experiments, we identify the optimal settings for adapting graph condensation techniques to multi-label scenarios, thus enhancing coherence and readability in the subsequent discussion.

### Issue 2: Optimization about Multi-Label Classification Loss.

In single-label settings, while the condensed graph is matched in various ways, the fixed label distribution can be used directly for synthetic graph initialization. With predefined synthetic label $Y'_{single} \in \{0, \cdots, C-1\}^{N'}$, the synthetic node feature is randomly retrieved from the particular class. As calculate the matching loss for nodes from different classes separately,is easier than that from all classes. In other words, this single-label class-wise sample method will directly sample each class to optimize. Specifically, for a given class $c$, it will sample a batch of nodes of class $c$ from original graph $\mathcal{G}$ with a portion of neighbors, denotes as $(A_c, X_c, Y_c) \sim \mathcal{G}$. For synthetic graph $S$, only sample the specific nodes of class $c$ without the neighbors. By doing so, the synthetic graph will use all the other nodes for aggregation stages during condensation, similarly, this process can be wrote as $(A'_c, X'_c, Y'_c) \sim \mathcal{S}$. Further in terms of details, for each class $c$, there is a subset of nodes $N_c$ that belong to this class. The loss is computed class-wise, for example, the cross-entropy loss for single node $j$ with ground truth label $y_j$ and predicted logit $z_{j,c}$ for class $c$ is:

$$\ell_{CE}(z_j, y_j) = -\log\left(\frac{e^{z_{j,y_j}}}{\sum_{c=0}^{C-1} e^{z_{j,c}}}\right) \tag{14}$$

The single-label final loss would be formulated as:

$$\mathcal{L}_{single-label} = \sum_{c=0}^{C-1} \sum_{j \in N_c} \ell_{CE}(z_j, y_j) \tag{15}$$

However, in multi-label classification $Y' \in \{y'_1, \cdots, y'_N\} \in \{0, 1\}^{N' \times K}$, each node can associate with multiple classes simultaneously. Typically, the multi-label classification task is modeled as $K$ independent binary

classification tasks, for each task $k$, the model predicts a logit $z_{j,k}$ and ground-truth label $y_{j,k}$ indicates whether node $j$ belongs to class $k$. In multi-label classification, SoftMarginLoss (Cao et al., 2019) incorporates margin-based optimization, which is beneficial when dealing with imbalanced datasets and overlapping classes. In multi-label settings, certain labels may frequently co-occur or exhibit correlations. SoftMarginLoss introduces a margin between positive and negative label predictions, encouraging the model to differentiate between these labels more confidently.

$$\mathcal{L}_{Softmargin}(z_{j,k}, y_{j,k}) = log(1 + e^{-z_{j,k} \cdot y_{j,k}}) \tag{16}$$

This formulation is particularly useful in ensuring that the model not only predicts correct labels but does so with confidence, reducing the chances of ambiguous predictions near the decision boundary. On the other hand, Binary Cross-Entropy Loss (Durand et al., 2019) is a widely-used objective function for multi-label classification, as it independently evaluates each label for each node in the graph. Then the task-wise loss for each binary classification task is computed using Binary Cross-Entropy (BCE) (Durand et al., 2019) loss:

$$\mathcal{L}_{\text{BCE}}(z_{j,k}, y_{j,k}) = -(y_{j,k} \log(\sigma(z_{j,k})) + (1 - y_{j,k}) \log(1 - \sigma(z_{j,k}))), \tag{17}$$

where $\sigma(z_{j,k}) = \frac{1}{1+e^{-z_{j,k}}}$ is the sigmoid function.

## 6.2 Experiments and Results Analysis

In this section, we investigate the effectiveness of various initialization and optimization strategies for multi-label graph condensation, aiming to determine the optimal settings. The results for characteristics of sampled subgraphs and additional metrics are provided in Appendix A. We conduct experiments using GCond to assess the impact of different initialization and optimization methods across representative datasets and their corresponding condensation rates. The condensation ratio (C-rate) is a critical measure in graph condensation, representing the fraction of the original graph retained in the condensed graph. For instance, with a C-rate of 1%, the synthetic graph $\mathcal{S}$ is only 1% of original graph $\mathcal{G}$. We then perform statistical analysis on the experimental results to compare the performance of different settings in multi-label graph condensation.

**Settings.** We select representative datasets for simplicity, using PPI as an example and testing its settings (with PPI-large excluded). The C-rate is determined based on the number of nodes in the synthetic graph. For instance, a 1.00% C-rate for PPI results in a synthetic graph of 147 nodes compared to the original 14,755 nodes. In general, synthetic graphs contain around 150 nodes. We evaluate different initialization strategies, including subgraph sampling methods (Random, Herding, K-Center), and probabilistic label sampling for synthetic $Y'$. After initialization, the synthetic graph $\mathcal{S}$ is optimized using the gradient matching strategy in GCond.

Next, we compare optimization loss functions, including SoftMarginLoss (Cao et al., 2019) and BCELoss (Durand et al., 2019), across various datasets with predefined condensation rates. These experiments aim to identify the optimal combination of initialization and loss function that balances graph size reduction and model performance in multi-label tasks.

**Results and Analysis.**

We evaluate the GCond framework under various multi-label settings, analyzing different initialization strategies and optimization methods with or without structure learning. Table 2 highlights F1-micro scores across subgraph and node-only settings, identifying strategies that best capture label distributions. Tables 3 and 4 compare optimization performance, while additional analysis examines the impact of structure information. Observing the experimental results, we draw three corresponding observation conclusions:

**Observation 1: K-Center sampling achieves optimal initialization.** K-Center sampling consistently delivers the highest F1-micro scores for dense, large-scale datasets (e.g., PPI, Yelp, OGBN-Proteins), as shown in Tables 2, 3, and 4. We believe K-Center sampling could cover the dataset's feature space more comprehensively by selecting a diverse set of nodes. This ensures better representation of all classes, including those that are underrepresented. In contrast, simpler strategies like Random and Herding, which focus less on diversity, perform better on sparser datasets (e.g., DBLP, PCG), where the feature space is less complex. Comparisons of SoftMarginLoss and BCELoss optimizations further reveal that K-Center initialization

achieves the best performance across loss functions, with or without graph structure learning. Notably, structure learning significantly enhances multi-label generalization in datasets like PPI and DBLP, showing its importance in improving performance.

**Observation 2: Preserving graph structure could improves performance of condensed large datasets.** Our analysis shows that incorporating graph structure information instead of relying solely on node attributes for label initialization leads to significant performance improvements, especially in large-scale datasets. This is because preserving the structural information enables a more accurate modeling of relationships and dependencies among nodes, crucial for maintaining multi-label interactions. For example, as shown in Table 2, using K-Center sampling on PPI with a condensation rate of 1.00%, preserving structure achieved an F1-micro score of 47.77%, compared to 36.60% when using nodes alone. This trend persists during the condensation process, where incorporating structure learning ($A'$) consistently enhances performance across various methods (see Tables 3 and 4). These findings underscore the critical role of structural information in improving the effectiveness of condensed graphs.

**Observation 3: BCELoss provides better overall results.** BCELoss consistently outperforms Soft-MarginLoss in terms of F1-micro scores, particularly when combined with structure learning. Theoretically, BCELoss could independently handle each label as a separate binary classification problem, thus better accommodating the non-mutually exclusive nature of labels in our case.For example, in the DBLP dataset with K-Center sampling and structure learning, BCELoss achieves 70.74%, compared to 60.77% with Soft-MarginLoss (Table 4). K-Center sampling also proves effective across dense datasets like PPI, Yelp, and OGBN-Proteins, while simpler methods such as Random and Herding perform relatively well on less dense datasets like DBLP and PCG but still fall short of K-Center. Preserving graph structure further enhances performance, as seen in the PPI dataset, where K-Center sampling achieves 47.77% with structure learning compared to 36.60% without it (Table 2). These patterns hold consistently across experiments, with BCELoss paired with K-Center initialization delivering the best results in most scenarios, regardless of structure learning. Notably, structure learning in advancing performance on datasets like PPI and DBLP, proves its effectiveness in condensation tasks. For additional comparisons and results, refer to Appendix A.

*Consistent with Observations 1 and 2, initializing with K-Center sampling and incorporating structure learning $A'$ could increase the performance of synthetic multi-label graph. As shown in Observation 3, arcoss a variety of datasets, BCELoss performs better than SoftMarginLoss. Particularly when paired with K-Center and structural learning, BCELoss improves the F1-micro score, suggesting it can offer greater label retention and general adaptability. Therefore, we adopt the K-Center and BCELoss as the initialization and optimization methods with structure information for multi-label scenario.*

## 7    Benchmarking

In this section, we adapt the discussed condensation methods to the multi-label scenario using the optimal settings outlined in Section 6. Table 5 compares coreset selection strategies (Random, Herding, K-Center) and graph condensation methods (SGDD, GCDM, GCond) against the full dataset at different condensation rates. SGDD and GCond utilize gradient matching between GNNs trained on the original graph $\mathcal{G}$ and the synthetic graph $\mathcal{S}$, while GCDM employs embedding matching based on nodes sharing the same class in the multi-label context. These results reveal key insights into the methods' effectiveness, dataset characteristics, and evaluation settings, with performance metrics reported as F1-scores in decimal values.

**Observation 1: GCond achieves the best results across datasets.** GCond consistently outperforms other methods on most datasets, particularly excelling in large-scale datasets with substantial label complexity. On the PPI dataset, GCond achieves F1-micro and F1-macro scores of 51.35% and 23.86%, respectively, at a 1.00% condensation rate. This robustness extends to PPI-large, where it achieves scores of 52.23% (F1-micro) and 28.59% (F1-macro) at a 0.40% condensation rate. Similarly, on DBLP, GCond demonstrates its superiority with F1-micro and F1-macro scores of 70.74% and 68.44%, respectively, at 0.80% condensation, showcasing its ability to preserve label correlations and structural integrity during condensation. However, GCond's advantage diminishes in the Yelp dataset at high condensation levels, likely due to the dataset's unique label structure, which poses challenges for methods relying on label coherence preservation.

Table 5: Graph Condensation Methods Performance Comparison. "-": out-of-memory (OOM) errors.

| Dataset | C-rate | Coreset (Random) | | Coreset (Herding) | | Coreset (K-Center) | | SGDD | | GCDM | | GCond | | Whole Dataset | |
|---|---|---|---|---|---|---|---|---|---|---|---|---|---|---|---|
| | | F1-micro | F1-macro | F1-micro | F1-macro | F1-micro | F1-macro | F1-micro | F1-macro | F1-micro | F1-macro | F1-micro | F1-macro | F1-micro | F1-macro |
| PPI | 0.50% | 41.92 | 14.69 | 41.16 | 12.76 | 45.65 | 15.54 | 44.99 | 22.58 | 48.28 | 23.37 | **49.63** | **23.16** | | |
| | 1.00% | 39.07 | 12.58 | 41.23 | 11.66 | 47.43 | 17.36 | 47.02 | 18.26 | 47.07 | 21.76 | **51.35** | **23.86** | 51.26 | 30.06 |
| | 2.00% | 41.12 | 14.12 | 39.21 | 10.23 | **52.34** | **26.62** | 46.73 | 21.24 | 47.91 | 21.93 | 51.21 | 24.32 | | |
| PPI-large | 0.10% | 44.43 | 14.49 | 43.17 | 13.74 | 41.12 | 18.11 | 49.32 | 18.33 | 47.13 | 37.59 | **51.05** | **27.51** | | |
| | 0.20% | 45.15 | 15.80 | 43.39 | 15.91 | 40.88 | 13.00 | 44.82 | 15.78 | 46.26 | 22.24 | **51.99** | **29.95** | 51.61 | 28.23 |
| | 0.40% | 44.01 | 15.43 | 43.61 | 13.74 | 41.04 | 12.89 | 49.89 | 29.08 | 44.08 | 19.48 | **52.23** | **28.59** | | |
| Yelp | 0.01% | 33.73 | 5.55 | 33.63 | 7.07 | **34.20** | **13.30** | - | - | 28.00 | 2.78 | 28.40 | 7.51 | | |
| | 0.02% | 33.64 | 5.09 | 32.94 | 5.64 | **35.64** | **12.30** | - | - | 27.84 | 3.76 | 33.94 | 6.22 | | |
| | 0.04% | 34.01 | 5.41 | 32.87 | 4.67 | **36.51** | **12.51** | - | - | 26.42 | 2.87 | 32.77 | 5.85 | 37.97 | 15.18 |
| | 0.05% | 33.74 | 5.86 | 32.83 | 5.17 | **36.63** | **11.30** | - | - | 24.10 | 2.42 | 34.11 | 6.86 | | |
| | 0.10% | 33.56 | 5.19 | 32.83 | 5.51 | **36.65** | **9.79** | - | - | 24.82 | 2.95 | 33.70 | 8.17 | | |
| | 0.20% | 33.75 | 5.50 | 33.20 | 4.71 | **37.61** | **11.48** | - | - | 28.43 | 3.32 | 32.41 | 12.18 | | |
| DBLP | 0.20% | 49.28 | 34.81 | 41.02 | 15.41 | 52.70 | 49.75 | 43.17 | 25.06 | 44.41 | 34.84 | **63.36** | **56.27** | | |
| | 0.40% | 52.94 | 44.71 | 42.59 | 31.05 | 55.27 | 50.89 | 45.37 | 43.42 | 45.88 | 26.54 | **68.11** | **57.35** | | |
| | 0.80% | 62.08 | 56.00 | 41.11 | 15.63 | 59.77 | 56.90 | 43.48 | 25.42 | 46.66 | 46.20 | **70.74** | **68.44** | 87.55 | 86.39 |
| | 1.60% | 68.39 | 63.75 | 42.24 | 25.78 | 63.87 | 62.34 | 42.73 | 34.22 | 45.99 | 26.47 | **70.49** | **68.92** | | |
| | 3.20% | 71.50 | 68.24 | 42.70 | 22.31 | 67.51 | 64.54 | 43.17 | 25.06 | 47.81 | 28.02 | **70.21** | **67.78** | | |
| OGBN-Proteins | 0.05% | 14.40 | 1.70 | 11.29 | 4.25 | **28.74** | **7.83** | - | - | 21.82 | 7.95 | 26.38 | 9.98 | | |
| | 0.10% | 15.63 | 2.12 | 15.28 | 5.21 | **31.04** | **10.13** | - | - | 22.18 | 3.07 | 29.40 | 7.47 | 10.14 | 9.16 |
| | 0.20% | 16.90 | 2.06 | 8.55 | 1.64 | 26.36 | 4.82 | - | - | 21.38 | 8.78 | **29.26** | **7.71** | | |
| PCG | 2.00% | 15.72 | 5.57 | 18.98 | 3.03 | 27.44 | 9.09 | 27.93 | 15.32 | **29.48** | **13.22** | 18.84 | 14.21 | | |
| | 4.00% | 15.83 | 5.20 | 23.03 | 13.84 | 25.13 | 5.73 | 28.40 | 9.00 | **32.16** | **8.31** | 25.58 | 13.32 | 42.26 | 31.49 |
| | 8.00% | 16.67 | 3.02 | 22.48 | 9.36 | 18.21 | 4.59 | **33.50** | **8.58** | 31.88 | 12.51 | 26.37 | 13.90 | | |
| HumanGo | 2.00% | 28.88 | 4.57 | 19.92 | 4.39 | 25.38 | 6.15 | 35.58 | 10.06 | **36.71** | **6.53** | 30.68 | 11.24 | | |
| | 4.00% | 27.95 | 7.98 | 31.70 | 6.66 | 31.94 | 7.24 | 35.39 | 9.19 | **35.79** | **7.31** | 34.91 | 10.74 | 51.67 | 25.57 |
| | 8.00% | **37.57** | **9.58** | 32.44 | 5.50 | 36.50 | 7.67 | 35.25 | 9.74 | 36.54 | 7.35 | 37.18 | 12.05 | | |
| EukaryoteGo | 1.00% | 31.45 | 5.57 | 21.60 | 4.24 | 17.79 | 2.66 | 29.40 | 3.25 | **36.93** | **4.09** | 35.24 | 5.57 | | |
| | 2.00% | 30.72 | 5.03 | 26.74 | 5.16 | 20.99 | 3.03 | 28.69 | 3.28 | **36.98** | **4.09** | 36.38 | 7.00 | 45.86 | 12.27 |
| | 3.00% | 30.79 | 4.18 | 25.99 | 3.43 | 22.08 | 4.31 | 36.57 | 4.75 | 36.97 | 4.84 | **38.90** | **6.04** | | |

**Observation 2: K-Center demonstrates stability in extreme condensation setting.** The K-Center method shows consistent performance, particularly on large-scale and dense-label datasets, highlighting its strength in preserving diversity at low condensation rates. On the Yelp dataset, K-Center outperforms traditional Coreset methods, achieving the highest F1-micro score (37.61%) at a 0.20% condensation rate. This suggests that its selection strategy effectively retains essential structural and label information, even under high condensation, making it advantageous for large-scale settings where extreme data reduction risks losing critical information.

**Observation 3: SGDD's performance is constrained by scalability issues on larger datasets.** SGDD faces OOM errors on large datasets like Yelp and OGBN-Proteins, revealing its scalability limitations. However, on smaller datasets, SGDD performs competitively, benefiting from its structure-broadcasting graphon technique, which efficiently captures structural patterns. For example, on datasets such as PCG, HumanGo, and EukaryoteGo, SGDD achieves strong F1-micro and F1-macro scores, demonstrating its effectiveness on smaller scales. Nevertheless, its applicability to larger datasets requires further optimization.

*While the performance of condensation methods varies with dataset characteristics, GCond excels on complex, high-dimensional datasets like DBLP and PPI, where maintaining label consistency and intricate relationships is crucial. In contrast, simpler methods such as Random and Herding show stable performance but are outperformed by GCond and K-Center on datasets that demand preservation of structure and label diversity. These results highlight the importance of selecting condensation methods based on dataset scale, complexity, and label interaction patterns to achieve optimal performance.*

## 8 Conclusion

In this benchmark, we introduce a comprehensive framework for multi-label graph condensation, extending traditional single-label methods to address the complexities of multi-label classification. Our results demonstrate that GCond not only preserves the original graph's structural integrity but also enhances label correlations, leading to more accurate multi-label predictions.

While our study establishes a baseline across standard benchmarks, the generalizability of these methods to other graph domains remains an open question. As a future direction, we aim to explore the applicability and robustness of our approach across diverse graph types, including dynamic graphs (Barros et al., 2021) (e.g., time-varying traffic flow graphs), heterogeneous graphs (Wang et al., 2022) (e.g., user-item graphs), and heterophilous graphs (Platonov et al., 2023) (e.g., graphs with dissimilar node labels). Such explorations will provide deeper insights into the flexibility and practical utility of our proposed methods across diverse real-world graph scenarios.

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

# A    Appendix

## A.1    Adapted GCond Algorithm

---

**Algorithm 1:** Multi-Label GCond Adaptation

---

**Input:** Training data $\mathcal{G} = (A, X, Y)$

1  Initialize multi-label synthetic graph $\mathcal{S}$ with $X', Y'$
2  **for** $k = 0$ **to** $K - 1$ **do**
3  $\quad$ Initialize $\theta_0 \sim P_{\theta_0}$
4  $\quad$ **for** $t = 0$ **to** $T - 1$ **do**
5  $\quad\quad$ $D' \leftarrow 0$
6  $\quad\quad$ Compute $A' = g_\phi(X')$, then $\mathcal{S} = (A', X', Y')$
7  $\quad\quad$ Compute $\mathcal{L}^{\mathcal{G}} = \mathcal{L}(GNN_{\theta_t^{\mathcal{G}}}(A, X), Y)$
8  $\quad\quad$ Compute $\mathcal{L}^{\mathcal{S}} = \mathcal{L}(GNN_{\theta_t^{\mathcal{S}}}(A', X', Y'))$
9  $\quad\quad$ $D' \leftarrow D' + D(\nabla_{\theta_t}\mathcal{L}^{\mathcal{G}}, \nabla_{\theta_t}\mathcal{L}^{\mathcal{S}})$
10 $\quad\quad$ **if** $t \% (\tau_1 + \tau_2) < \tau_1$ **then**
11 $\quad\quad\quad$ Update $X' \leftarrow X' - \eta_1 \nabla_{X'} D'$
12 $\quad\quad$ **else**
13 $\quad\quad\quad$ Update $\phi \leftarrow \phi - \eta_2 \nabla_\phi D'$
14 $\quad\quad$ Update $\theta_{t+1} \leftarrow \text{opt}_\theta(\theta_t, \mathcal{S}, \tau_\theta)$
15 $\quad$ Compute new adjacency matrix: $A' = g_\phi(X')$
16 $\quad$ **for** *each edge* $(i, j)$ *in* $A'$ **do**
17 $\quad\quad$ $A'_{i,j} \leftarrow A'_{i,j}$ if $A'_{i,j} > \delta$, otherwise $A'_{i,j} \leftarrow 0$
18 **return** $(A', X', Y')$

---

Table 6: Graph Condensation Methods Performance Comparison on Multi-label Benchmarks. "-": Out-of-memory (OOM); "*": Estimated via smaller batch training due to memory constraints.

| Dataset | C-rate | Coreset (Random) | | Coreset (Herding) | | Coreset (K-Center) | | SGDD | | GCDM | | GCond | | GC-SNTK* | |
|---|---|---|---|---|---|---|---|---|---|---|---|---|---|---|---|
| | | F1-Micro | F1-Macro | F1-Micro | F1-Macro | F1-Micro | F1-Macro | F1-Micro | F1-Macro | F1-Micro | F1-Macro | F1-Micro | F1-Macro | F1-Micro | F1-Macro |
| PPI | 0.50% | 41.92 | 14.69 | 41.16 | 12.76 | 45.65 | 15.54 | 44.99 | 22.58 | 48.28 | 23.37 | **49.63** | 23.16 | 38.76 | 13.25 |
| | 1.00% | 39.07 | 12.58 | 41.23 | 11.66 | 47.43 | 17.36 | 47.02 | 18.26 | 47.07 | 21.76 | **51.35** | **23.86** | 41.47 | 15.90 |
| | 2.00% | 41.12 | 14.12 | 39.21 | 10.23 | **52.34** | **26.62** | 46.73 | 21.24 | 47.91 | 21.93 | 51.21 | 24.32 | 45.25 | 16.94 |
| PPI-large | 0.10% | 44.43 | 14.49 | 43.17 | 13.74 | 41.12 | 18.11 | 49.32 | 18.33 | 47.13 | 37.59 | **51.05** | 27.51 | - | - |
| | 0.20% | 45.15 | 15.80 | 43.39 | 15.91 | 40.88 | 13.00 | 44.82 | 15.78 | 46.26 | 22.24 | **51.99** | 29.95 | - | - |
| | 0.40% | 44.01 | 15.43 | 43.61 | 13.74 | 41.04 | 12.89 | 49.89 | 29.08 | 44.08 | 19.48 | **52.23** | 28.59 | - | - |

## A.2 Recent Condensation Methods Discussion

Recent condensation methods have investigated various strategies to deal with the challenges in scalability, robustness, inductiveness, and explainability. While selecting methods for our benchmark, we mainly aimed for fair and systematic evaluation across a consistent condensation setting. Several recent methods fall outside this scope or present practical issues that prevent their direct inclusion or fair comparison. Here we will summarize and discuss these methods, categorizing them by the considerations leading to their exclusion or special treatment in our current benchmark.

**Dynamic and Open-world Settings.** The Open-world Graph Condensation (OpenGC) Gao et al. (2024) solves distribution shifts in dynamic graph settings through structure-aware condensation of evolving graphs. It utilizes kernel ridge regression and non-parametric graph convolutions to update condensed graphs for life-long learning scenarios. While our benchmark focuses primarily on graph condensation scenarios under a static distribution setting common in multi-label graph classification. Meanwhile, at the time of our study, the implementation code was not publicly available yet. Future extensions of our work could integrate dynamic and evolving datasets, providing an evaluation benchmark suited explicitly to assess OpenGC and similar dynamic methods.

**Interpretability Constraints.** The EXGC (Efficient and eXplainable Graph Condensation) Fang et al. (2024) method proposes variational approximations coupled with Gradient Information Bottlenecks (GDIB) to enhance the interpretability in large-scale graph condensation tasks. Due to the absence of publicly accessible implementations at the time of our benchmark, EXGC could not be integrated despite its potential practical and interpretability advantages. With code availability, future benchmarks may consider evaluating EXGC's effectiveness.

**Inductive Learning Scenarios.** The Mapping-aware Graph Condensation (MCond) Gao et al. (2024) learns node mappings between the original large graph and the synthetic condensed graph from an inductive learning perspective. It allows integration of unseen (inductive) nodes without referring back to the original large graph at inference time. However, our study strictly evaluates methods under the transductive settings where nodes are fully observed during training. Since MCond targets inductive and unseen node inference specifically, its inductive-focused design places it beyond our present evaluation objectives. Expanding our benchmark to explicitly include inductive graph condensation paradigms remains a prospective focus for future work.

**Practical Implementation Limitations.** The Graph Condensation via Structure-based Neural Tangent Kernel (GC-SNTK) Wang et al. (2024) uses structure-informed neural tangent kernels to complete the condensation stage without bi-level constraints. In our experimentation we encouter several challenges: (1) its kernel computation is limited to training set nodes, unlike methods utilizing entire graph structures; (2) it struggles with complex label dependencies, as it optimizes synthetic node features $A'$ and labels $Y'$ in a single-label setting; (3) memory limitations due to kernel scalability issues, which restrict batch sizes and may lead to unfair comparisons under standardized benchmarking conditions; (4) a sparse structure requirement for kernel optimization, which is problematic as multilabel graphs have denser edges and more complex structures. Despite these challenges, we implemented GC-SNTK in a simplified form in our code https://github.com/liangliang6v6/Multi-GC. Sampled results are illustrated by the PPI dataset in Table 6.

Adjustments such as kernel generalization and more sophisticated label-aware formulations are necessary to effectively leverage GC-SNTK for multilabel tasks.

## A.3 Datasets and Metrics

**Datasets.** We use eight real-world datasets from diverse domains, including bioinformatics and social networks, to assess the efficacy of graph condensation methods in multi-label scenarios. A detailed description of each dataset is listed as follows:

- PPI (Zeng et al., 2019) is the Protein-Protein Interaction network. It classifies protein functions based on the interactions of human tissue proteins. Positional gene sets are used, motif gene sets and immunological signatures as features and gene ontology sets as labels (121 in total), collected from the Molecular Signatures Database (Liberzon et al., 2015). And PPI-large is the larger version of PPI dataset.

- Yelp (Zeng et al., 2019) is processed from a public open dataset from yelp.com. The nodes represent active users and the edges are the relationship between the users. The multi-label of each node represents the types of business like Coffee & Tea, Flowers & Gifts, Tours, and so on. Node features contain the information of all the reviews and rates by users, generated by pre-trained Word2Vec model (Church, 2017).

- DBLP (Akujuobi et al., 2019) is the citation network extracted from DBLP. The nodes represent authors and edges are the co-authorship between the authors. Multi-label here indicates the four research areas like database, data mining, information and retrieval and artificial intelligence.

- OGBN-Proteins from Open Graph Benchmark (Hu et al., 2020) is an undirected, weighted, and typed (according to species) graph. Nodes represent proteins, and edges indicate different types of biologically meaningful associations between proteins, e.g., physical interactions, co-expression or homology. The task is to predict the presence of protein functions in a multi-label binary classification setup, where there are 112 kinds of protein functions denote as labels to predict in total. Here the protein functions refer to the specific biological activities or roles that a protein performs within a cell or organism. These functions are typically described using Gene Ontology (GO) database (Huntley et al., 2015) or other biological annotation systems and can fall into different categories.

- PCG (Zhao et al., 2023) is Protein-Phenotype graph dataset, which focused on predicting phenotypes associated with proteins. Phenotypes refer to observable traits or characteristics of diseases, and identifying these associations can be valuable for clinical diagnostics or discovering potential drug targets. The dataset is constructed using protein-phenotype associations from the DisGeNET database (Piñero et al., 2020), where each protein is linked to one or more phenotypes. The phenotypes are grouped into disease categories based on the MESH ontology (Bhattacharya et al., 2011), and labels with fewer than 100 associated proteins are removed. The PCG includes 3,233 proteins as nodes and 37,351 protein-protein interactions as edges. The task is to predict multiple phenotype labels for each protein based on its interactions and features.

- HumanGo (Chou & Shen, 2007; Liberzon et al., 2015) contains 3,106 proteins, each potentially associated with one or more of 14 subcellular locations as multi-label. The proteins are represented as nodes in a graph, with features generated from their sequences using pre-trained model (Yang et al., 2020). Protein-protein interactions form the edges of the graph, with each edge representing the confidence of different interaction types. Different location associations come from the GO database (Huntley et al., 2015), which assigns standardized terms to proteins based on their roles in biological processes (e.g., cell division, metabolism), molecular functions (e.g., enzyme activity, receptor binding), and cellular components (e.g., nucleus, membrane).

- EukaryoteGo (Chou & Shen, 2007; Liberzon et al., 2015) follows a similar structure but focuses on eukaryote proteins. It includes 7,766 proteins and their interactions, with proteins being assigned to one or more of 22 subcellular locations. Like HumanGo, it uses protein sequences for node features and protein-protein interactions for graph structure.

**Metrics.** To evaluate the performance of graph condensation methods in multi-label classification, we use two widely recognized metrics: F1-micro and F1-macro (Lipton et al., 2014):

- F1-micro score aggregates the contributions of all classes to compute the average F1 score. It does this by first calculating the total true positives, false positives, and false negatives across all labels, and then using these aggregated values to compute a global precision and recall. The F1-micro score is particularly useful when dealing with imbalanced datasets, as it gives equal weight to each instance rather than each class. It is defined as:

$$\text{F1-micro} = \frac{2 \times \text{Precision} \times \text{Recall}}{\text{Precision} + \text{Recall}} \tag{18}$$

  where:

$$\text{Precision} = \frac{\text{TP}}{\text{TP} + \text{FP}}$$
$$\text{Recall} = \frac{\text{TP}}{\text{TP} + \text{FN}}$$

- F1-macro score computes the F1 score for each class independently and then averages these scores. This metric treats all classes equally, regardless of their frequency in the dataset. It is particularly useful for understanding the model's performance across all classes, especially in scenarios where some classes may be underrepresented. The F1-macro score is defined as:

$$\text{F1-macro} = \frac{1}{C} \sum_{i=1}^{C} \text{F1}_i \tag{19}$$

  where $C$ is the number of classes and $\text{F1}_i$ is the F1 score for the $i^{th}$ class, calculated as:

$$\text{F1}_i = \frac{2 \times \text{Precision}_i \times \text{Recall}_i}{\text{Precision}_i + \text{Recall}_i} \tag{20}$$

### A.4 Visualization of Multi-Label Datasets

In this section, we delve deeper into the characteristics of the condensed graphs by analyzing class distribution and label correlation within the multi-label datasets. This detailed examination is crucial for understanding how effectively the condensation methods preserve the multi-label relationships present in the original datasets. After obtaining the full benchmark results, we provide a detailed analysis of the condensed graphs by examining the class distribution and label correlation within the synthetic graphs. These statistical properties are crucial for understanding how well the condensed graphs preserve the multi-label relationships present in the original datasets.

- Label Correlation: To analyze the relationships between labels in the dataset, we define the correlation matrix: Let $G_{\text{labels}}$ be the set of labels from the original dataset $\mathcal{G}$, and $Y$ be the corresponding original label matrix. We denote $M(i, j)$ as the occurrence times of labels $L_i$ and $L_j$, and $N_i$ as the total number of times label $L_i$ appears.

  The conditional probability $P(i, j)$ is defined as:

$$P(i, j) = \frac{M(i, j)}{N_i} \tag{21}$$

  This conditional probability matrix $P$ models the edges as a co-occurrence matrix.

  Next, we define the diagonal matrix $D$ as follows:

$$D = \text{Diag}(P) \tag{22}$$

Finally, the Laplacian matrix $L_o$ is computed as:

$$L_o = D - P \tag{23}$$

By computing the label correlation matrix, we investigate whether the co-occurrence patterns between labels in the original graph are retained in the condensed graph.

- Class Distribution: We analyze how evenly or unevenly the labels are distributed across the classes in different datasets by index, which helps assess the quality of the condensation process.

Following the above definition, we report the visualizations of different datasets as follows. The label correlation and class distribution are shown in Figures 2 and 3, respectively. We find that the more complex the labels, the more the condensation methods would rely on structure and become more random. For future work, more suitable methods need to be fit to multi-label scenarios.

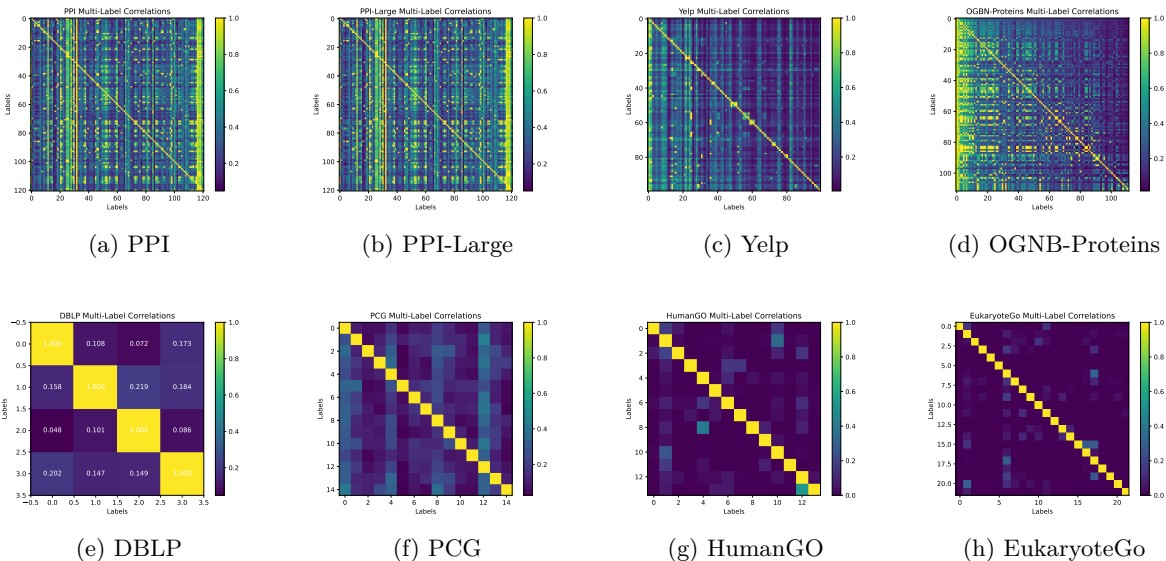

Figure 2: Multi-label Correlation Visualization

## A.5 Characteristics of Subgraph Initialization

Analyzing subgraph structures requires characterizing two essential aspects: structural connectivity and attribute-driven node similarity (homophily). Connectivity captures how tightly connected the subgraph is, quantified by the average connected component size:

$$C = \frac{1}{m} \sum_{i=1}^{m} |C_i|, \tag{24}$$

where $C_i$ represents each connected component and $m$ is the total number of components.

On the other hand, homophily measures how interconnected nodes share similar attributes, especially relevant in multi-label scenarios. Using binary attribute vectors $y_u$ and the Jaccard similarity, subgraph homophily $H$ is defined as the average pairwise similarity across edges:

$$H = \frac{1}{|E|} \sum_{(u,v) \in E} J(y_u, y_v), \quad \text{with} \quad J(y_u, y_v) = \frac{|y_u \cap y_v|}{|y_u \cup y_v|}. \tag{25}$$

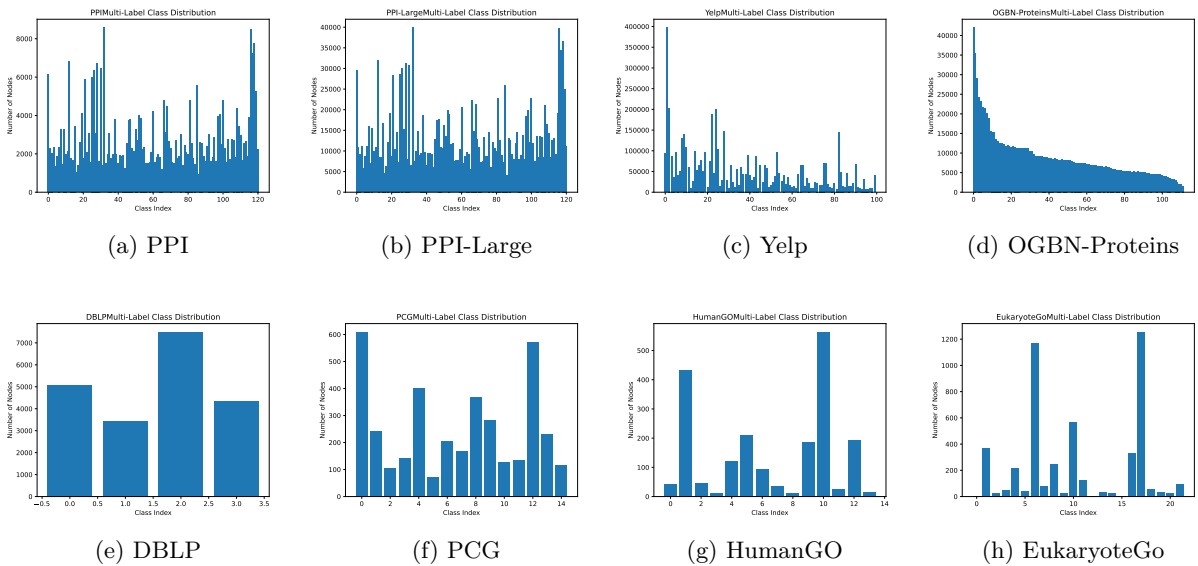

Figure 3: Multi-label Class Distribution Visualization

Based on subgraph connectivity (Table 7) and homophily (Table 8), we observe that methods preserving higher structural and attribute-based similarity consistently achieve better F1 performance (Table 2). For instance, in datasets like PPI and OGBN-Proteins, the K-Center method maintains higher subgraph connectivity and homophily compared to Random and Herding, resulting in clearly superior predictive accuracy. Conversely, for datasets such as PCG, the Herding method's better homophily and connectivity align with its improved classification performance. These outcomes highlight that maintaining structural coherence and attribute similarity in subgraphs directly supports better multi-label classification results.

| Datasets | C-rate | Connectivity | | | Whole Dataset |
|---|---|---|---|---|---|
| | | Random | Herding | K-Center | |
| PPI | 1.00% | 24.5 | 17.8 | **32.6** | 216.98 |
| Yelp | 0.02% | 9.7 | 12.2 | **18.3** | 114 |
| DBLP | 0.80% | 5.6 | 3.8 | **6.9** | 7.93 |
| OGBN-Proteins | 0.10% | 55.4 | 46.2 | **79.8** | 295.45 |
| PCG | 4.00% | 24.3 | **42.2** | 36.7 | 129.32 |

Table 7: Subgraph Connectivity Characteristics Across Different Methods with Condensation Rates

| Datasets | C-rate | Homophily | | | Whole Dataset |
|---|---|---|---|---|---|
| | | Random | Herding | K-Center | |
| PPI | 1.00% | 0.24 | 0.22 | **0.30** | 0.36 |
| Yelp | 0.02% | 0.15 | **0.21** | 0.19 | 0.26 |
| DBLP | 0.80% | 0.60 | 0.55 | **0.69** | 0.76 |
| OGBN-Proteins | 0.10% | 0.33 | 0.29 | **0.45** | 0.53 |
| PCG | 4.00% | 0.10 | **0.15** | 0.13 | 0.17 |

Table 8: Subgraph Homophily Characteristics Across Different Methods with Condensation Rates

## A.6 F1-Macro Results of Adaptation

Figures 4, 5 and 6 show the visualizations of different adapation strategies measured by F1-micro score. To further investigate the effectiveness of the methods, we also report the F1-macro score in Tables 9 , 10 and 11.

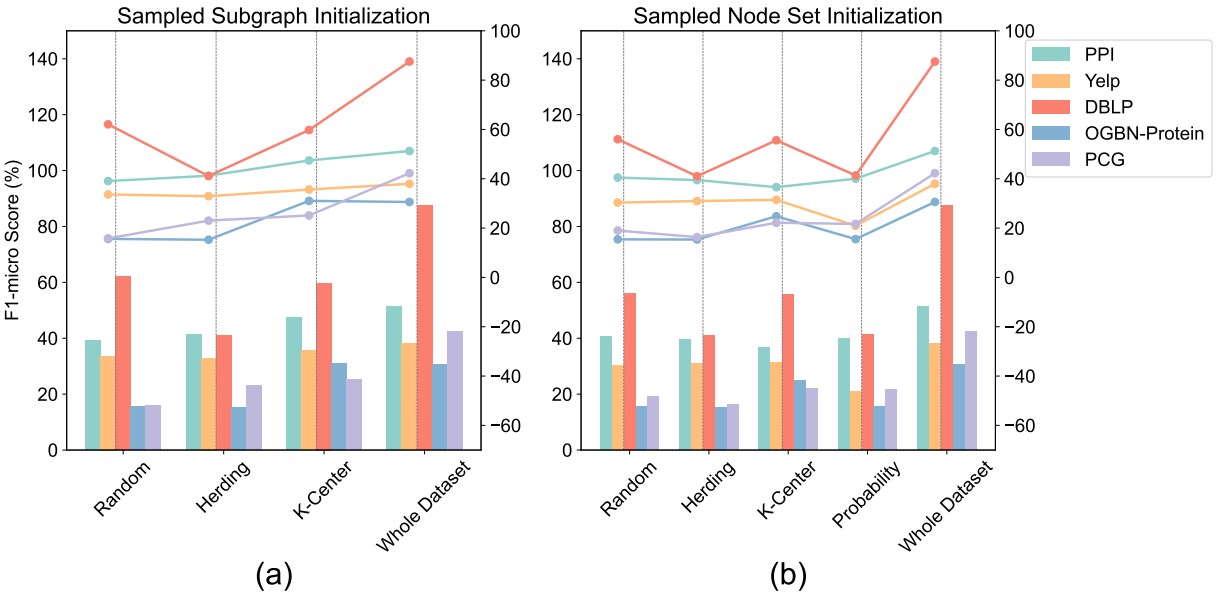

Figure 4: Different Initialization Methods Performance. Performance metrics of the model, with the F1-score represented as a decimal value.

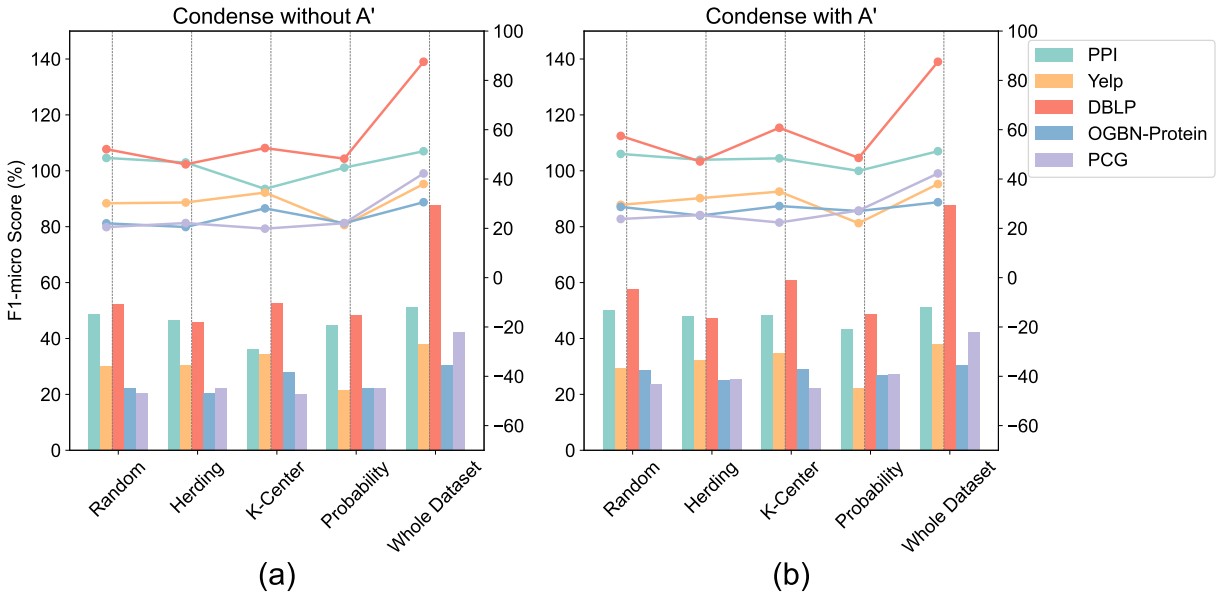

Figure 5: Performance Optimized by SoftMargin Loss. Performance metrics of the model, with the F1-score represented as a decimal value.

## A.7 Class-Weighted Optimization

Let $N_{\max} = \max_{c \in 0, \cdots, C-1} N_c$ is the maximum number of samples across all classes. For each class $c$ the class-wise coefficient is defined as:

$$\alpha_c = \frac{N_c}{N_{\max}} \tag{26}$$

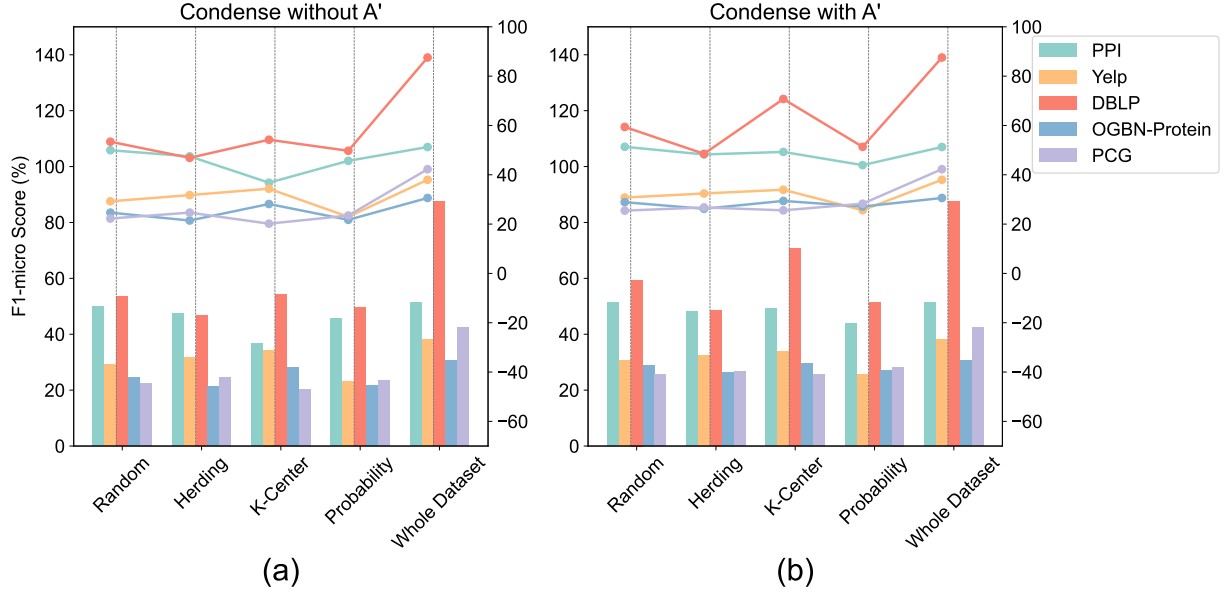

Figure 6: Performance Optimized by BCE Loss. Performance metrics of the model, with the F1-score represented as a decimal value.

| Datasets | C-rate | Random | | Herding | | K-Center | | Probability | Whole Dataset |
|---|---|---|---|---|---|---|---|---|---|
| | | Subgraph | Nodes | Subgraph | Nodes | Subgraph | Nodes | | |
| **PPI** | **1.00%** | 12.58 | 10.67 | 11.66 | 17.98 | 17.36 | **18.37** | 12.55 | 30.06 |
| **Yelp** | **0.02%** | 5.09 | 3.76 | 5.64 | 3.66 | **12.30** | 8.78 | 2.87 | 15.18 |
| **DBLP** | **0.80%** | 56.00 | 51.43 | 15.63 | 15.41 | **56.90** | 52.74 | 15.68 | 86.39 |
| **OGBN-Pro** | **0.1%** | 2.12 | 7.12 | 5.21 | 7.17 | **10.13** | 6.42 | 6.36 | 9.16 |
| **PCG** | **4%** | 5.20 | 9.69 | **13.84** | 6.45 | 5.73 | 7.10 | 11.35 | 31.49 |

Table 9: F1 Macro Score(%) of Coreset Method with Different Initializetion Strategies

The total loss for all the $N$ nodes would be the sum over all the classes and their respective nodes $N_c$:

$$\mathcal{L}_{single-label} = \sum_{c=0}^{C-1} \alpha_c \sum_{j \in N_c} \ell_{CE}(z_j, y_j) \tag{27}$$

| Datasets | C-rate | Random | | Herding | | K-Center | | Probability | | Whole Dataset |
|---|---|---|---|---|---|---|---|---|---|---|
| | | Without $A'$ | With $A'$ | Without $A'$ | With $A'$ | Without $A'$ | With $A'$ | Without $A'$ | With $A'$ | |
| **PPI** | **1.00%** | 22.60 | 28.33 | **30.37** | 18.32 | 11.62 | 23.12 | 16.66 | 12.61 | 30.06 |
| **Yelp** | **0.02%** | 3.72 | 6.98 | 3.57 | 3.99 | 6.37 | **9.02** | 2.13 | 1.86 | 15.18 |
| **DBLP** | **1%** | 35.81 | 37.84 | 27.08 | 27.11 | 32.21 | **49.36** | 42.33 | 38.08 | 86.39 |
| **OGBN-Pro** | **0.10%** | 4.36 | 5.47 | 5.37 | **7.18** | 5.26 | 5.32 | 3.21 | 4.76 | 9.16 |
| **PCG** | **4%** | 12.42 | 13.86 | 10.09 | **15.50** | 7.34 | 9.47 | 13.29 | 6.50 | 31.49 |

Table 10: F1-Macro Score (%) of GCond Method with Random/Herding/K-Center/Probability Distribution Initialization with/without Learning from Graph Structure for SoftMarginLoss

| Datasets | C-rate | Random | | Herding | | K-Center | | Probability | | Whole Dataset |
|---|---|---|---|---|---|---|---|---|---|---|
| | | Without $A'$ | With $A'$ | Without $A'$ | With $A'$ | Without $A'$ | With $A'$ | Without $A'$ | With $A'$ | |
| **PPI** | **1.00%** | 22.82 | **23.86** | 21.72 | 21.35 | 20.69 | 21.47 | 17.68 | 20.62 | 30.06 |
| **Yelp** | **0.02%** | 3.57 | **8.02** | 4.24 | 4.13 | 6.13 | 6.22 | 4.00 | 3.07 | 15.18 |
| **DBLP** | **1%** | 34.89 | 51.44 | 26.94 | 28.28 | 34.87 | **68.44** | 38.36 | 39.87 | 86.39 |
| **OGBN-Pro** | **0.10%** | 6.79 | 6.19 | 5.87 | 7.30 | 4.98 | **7.47** | 2.89 | 4.99 | 9.16 |
| **PCG** | **4%** | 7.83 | 7.29 | **15.17** | 14.27 | 8.02 | 13.32 | 13.13 | 11.30 | 31.49 |

Table 11: F1-Macro Score (%) of GCond Method with Random/Herding/K-Center/Probability Distribution Initialization with/without Learning from Adjacent Nodes for BCELoss

| Datasets | C-rate | Random | | Herding | | K-Center | | Probability | | Whole Dataset |
|---|---|---|---|---|---|---|---|---|---|---|
| | | Without $A'$ | With $A'$ | Without $A'$ | With $A'$ | Without $A'$ | With $A'$ | Without $A'$ | With $A'$ | |
| **PPI** | **1.00%** | 49.30 | **51.28** | 46.25 | 46.65 | 48.10 | 48.15 | 46.32 | 43.64 | 51.26 |
| **Yelp** | **0.02%** | 31.06 | 30.12 | 32.42 | 32.58 | **34.26** | 33.75 | 22.10 | 25.82 | 37.97 |
| **DBLP** | **0.80%** | 52.47 | **65.27** | 46.19 | 48.79 | 52.58 | 60.09 | 50.43 | 52.69 | 87.55 |
| **OGBN-Proteins** | **0.10%** | 26.51 | 23.25 | 22.25 | 26.36 | 24.81 | 28.24 | 26.47 | **30.13** | 30.59 |
| **PCG** | **4%** | 23.78 | 25.72 | 18.36 | 23.53 | 21.15 | 24.10 | 24.68 | **30.13** | 42.26 |

Table 12: F1-Micro Score (%) of GCond Method with Random/Herding/K-Center/Probability Distribution Initialization with/without Learning from Adjacent Nodes for BCE Loss and Balanced Coefficient

This is the general equation for different methods, as the condensation part is using the $\mathcal{M}(\cdot)$ as the matching loss, for example, GCond would compute the gradient for each class with class-wise coefficient and sum them up to get the final loss.

Furthermore, for each task we are working on the binary classification problem. To better capture the contribution of each class in multi-label, we introduce class weights $\omega \in \mathbb{R}^K$ for each task $k$:

$$\ell_{\text{BCE}}(z_{j,k}, y_{j,k}) = -\omega_k(y_{j,k}\log(\sigma(z_{j,k})) + (1 - y_{j,k})\log(\sigma(z_{j,k}))), \tag{28}$$

where $\omega_k$ is the positive class weight, scaling the loss for each positive class $y_{i,k} = 1$. Therefore, the final loss of multi-label classification is:

$$\mathcal{L}_{multi-label} = -\frac{1}{N}\sum_{j=0}^{N-1}\sum_{k=0}^{K-1}\omega_k(y_{j,k}\log(\sigma(z_{j,k})) \\ +(1 - y_{j,k})\log(1 - \sigma(z_{j,k}))) \tag{29}$$

This is helpful when there is an imbalance between the number of positive and negative samples. The F1-micro results are shown in Table 12 and Figure 7. F1-macro results are shown in Table 13.

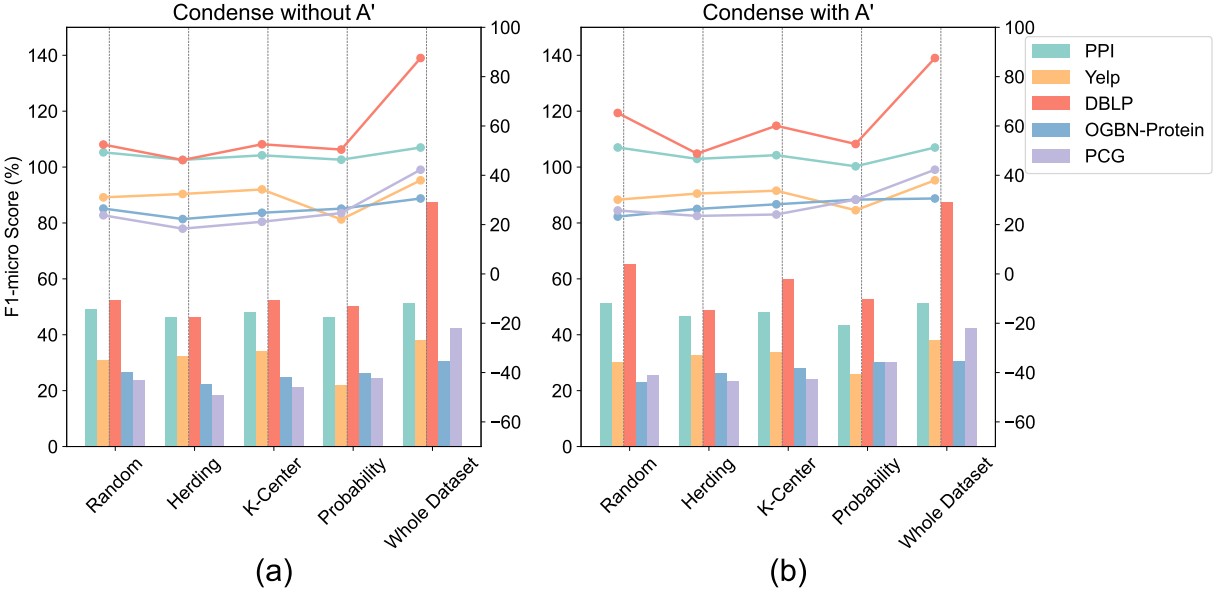

Figure 7: Performance Optimized by BCE Loss with Class Weight $\omega$

| Datasets | C-rate | Random | | Herding | | K-Center | | Probability | | Whole Dataset |
|---|---|---|---|---|---|---|---|---|---|---|
| | | Without $A'$ | With $A'$ | Without $A'$ | With $A'$ | Without $A'$ | With $A'$ | Without $A'$ | With $A'$ | |
| PPI | 1.00% | 21.75 | 25.65 | 15.55 | 27.21 | 27.09 | **31.97** | 14.31 | 21.43 | 30.06 |
| Yelp | 0.02% | 3.91 | 6.49 | 4.06 | 5.16 | **11.72** | 7.78 | 1.86 | 2.42 | 15.18 |
| DBLP | 0.80% | 32.79 | **61.83** | 26.44 | 29.25 | 32.64 | 56.29 | 37.78 | 42.11 | 86.39 |
| OGBN-Pro | 0.10% | 4.60 | 6.05 | 3.54 | 4.69 | 6.42 | 7.89 | 4.38 | **8.56** | 9.16 |
| PCG | 4% | 13.26 | 11.93 | 12.24 | **15.75** | 5.70 | 9.88 | 11.60 | 8.56 | 31.49 |

Table 13: F1-Macro Score (%) of GCond Method with Random/Herding/K-Center/Probability Distribution Initialization with/without Learning from Adjacent Nodes for BCE Loss and Balanced Coefficient

