# OpenReview forum: "Extending Graph Condensation to Multi-Label Datasets: A Benchmark Study"
_TMLR — Accepted by TMLR_

### Review · Reviewer_2yYV · 2025-02-03

**Summary Of Contributions:**

This paper is a benchmarking study for off-the-shelf graph condensation methods to the multi-label setting. The main contribution is to show the behavior of each extended graph condensation methods under extensive experiments.

**Audience:**

No

**Claims And Evidence:**

Yes

**Requested Changes:**

1. Please address the issue on the motivation of graph condensation that I mentioned above.
2. Please correct the reference format.

**Strengths And Weaknesses:**

## Strengths

(+) The setting of multi-label graph condensation is new.

(+) The experiment seems extensive.

## Weaknesses

(-) The current writing, especially about the motivation on why graph condensation is meaningful, relies on readers to agree with prior graph condensation works in advance.

(-) The citation format is pretty off. It seems like the authors recycle the paper from other venue format but did not even bother to check and proofread.

## Neutral

(*)  I know the TMLR policy does not care about novelty, but I want to say the proposed multi-label extension is not as non-trivial as the authors’ claim in the introduction.

## Detail comments

First of all, I admit that I am not familiar with the line of works on graph condensation so I may misunderstand something here. Please correct me if I am wrong. According to the authors introduction and the section about graph condensation, the goal is to learn a synthetic graph so that the GNN trained on this graph will be similar as if it is trained on the original training graph. The motivation of graph condensation is that training on this small synthetic graph is much more efficient. However, what I do not understand is that the process of obtaining this synthetic graph is already more costly than training on the original graph directly. Thus, unless the condensed graph can be used for inference, it makes no sense to me. Unfortunately, according to Figure 1, we can see that even the number of nodes are different, which implies that it is hard (or even impossible) to do inference tasks using the condensed graph unless some mapping can be established. I feel the prior graph condensation works must have addressed this issue and I feel the authors should address (or at least mentioned and referred to prior works) it. The paper should by default be self-contained.

Another thing is that the citation format of the paper is quite off. One should use \citet{} and \citep{} properly. Right now, there is even no space between the name from the citations and the main text, which make the paper hard to read. I feel this issue is very obvious and can be avoided by a brief proof read. Finally, while TMLR does not care about novelty, but I want to say the proposed multi-label extension is not as non-trivial as the authors’ claim in the introduction. It is nothing wrong with this point, but I just want to state it as a fact to help the judgement of AC.

---

> ### Author Response · Authors · 2025-03-07
>
> Thank you for the comments. Indeed, graph condensation generally involves an initial computational overhead higher than training once on the original graph but becomes highly beneficial in repeated training scenarios, including hyperparameter optimization and neural architecture search (NAS). In these use cases, models are trained multiple times. Therefore, condensed graphs substantially accelerate experimentation due to their reduced size and computational costs. Regarding inference scenarios, the reviewer is correct that condensed graphs typically differ in node counts and identities from the original graphs, complicating direct inference tasks like transductive node classification without additional mapping steps. Existing graph condensation methods primarily target inductive settings, where condensed graphs serve as representative training sets, allowing generalization to unseen nodes or graphs. The direct node-level inference would require further mapping techniques or additional refinements; exploring these remains an open area for future research.
>
> We apologize for the incorrect and inconsistent citation formatting, and will carefully proofread the manuscript to ensure proper usage of commands throughout. This revision will correct the spacing issues and improve overall readability and clarity. We also appreciate the reviewer's transparency regarding novelty. Our primary contribution is a systematic benchmarking study that extends existing single-label graph condensation methods to the multi-label setting. While we agree that the extension itself may not be highly complex, we believe our comprehensive empirical evaluation and clear methodological framework provide valuable insights into applying graph condensation for multi-label scenarios.

---

### Review · Reviewer_KUjt · 2025-02-12

**Summary Of Contributions:**

As a benchmark-type study, this paper extends existing graph condensation techniques from single-label classification task on graphs to multi-label datasets, making a significant contribution to the field of graph condensation. The empirical results are extensive and compelling, and the insights provided are valuable for researchers. However, the paper lacks in-depth theoretical analysis and a more comprehensive baseline comparison. With these improvements, the paper would be even more complete.

**Audience:**

Yes

**Claims And Evidence:**

Yes

**Requested Changes:**

Cons:

1.	While the paper provides extensive empirical results, it lacks a theoretical analysis of why certain methods (e.g., K-Center initialization, BCELoss) perform better in multi-label settings. A deeper theoretical justification could strengthen the paper's contributions.

2.	The paper compares the proposed methods against each other but does not include a comparison with state-of-the-art methods [1-4].

3.	While the paper evaluates the methods on several datasets, it would be beneficial to discuss how well the proposed techniques generalize to other domains or types of graph data (e.g., dynamic graphs, heterogeneous graphs, heterophilous graphs). This could be a direction for future work.

4.	Including a link to the code or a more detailed appendix with implementation details would improve the reproducibility of the results.

[1] Gao X, Chen T, Zang Y, et al. Graph condensation for inductive node representation learning[C]//2024 IEEE 40th International Conference on Data Engineering (ICDE). IEEE, 2024: 3056-3069.
[2] Fang J, Li X, Sui Y, et al. Exgc: Bridging efficiency and explainability in graph condensation[C]//Proceedings of the ACM on Web Conference 2024. 2024: 721-732.
[3] Wang L, Fan W, Li J, et al. Fast graph condensation with structure-based neural tangent kernel[C]//Proceedings of the ACM on Web Conference 2024. 2024: 4439-4448.
[4] Gao X, Chen T, Zhang W, et al. Graph condensation for open-world graph learning[C]//Proceedings of the 30th ACM SIGKDD Conference on Knowledge Discovery and Data Mining. 2024: 851-862.

**Strengths And Weaknesses:**

Pros:

1.	The paper is well-structured, with a clear problem formulation, detailed methodology, and comprehensive experimental results. It introduces several adaptations to existing graph condensation methods to handle multi-label datasets. The modifications to synthetic dataset initialization and optimization strategies are well-motivated and clearly explained.

2.	The authors conduct extensive experiments on eight real-world multi-label datasets, providing a thorough evaluation of their proposed methods. The use of multiple datasets from diverse domains (e.g., PPI, Yelp, DBLP) enhances the generalizability of the findings.

3.	The paper provides three key observations based on the experimental results:
- K-Center sampling is optimal for initialization.
- Preserving graph structure improves performance, especially for large datasets.
- Binary Cross-Entropy Loss (BCELoss) generally outperforms SoftMarginLoss.
   These insights are valuable for researchers working on graph condensation.

---

> ### Author Response · Authors · 2025-03-07
>
> 1. Thank you for the suggestion. We have provided additional theoretical insights into why K-Center initialization and BCELoss perform better in multi-label settings in Section 6.1. K-Center selects representative nodes that cover the feature space more effectively. Similarly, BCELoss is well-suited for multi-label scenarios as it independently models each label as a separate binary classification task, making it more adaptable to label overlap and imbalances.
> 2. We appreciate the suggestion to include additional comparisons with state-of-the-art methods. We are actively working on incorporating these evaluations. However, due to the unavailability of official implementations for [1], [2], and [4], we are unable to include them at this moment. We will integrate these methods into our benchmark once their code is released. For a fair comparison, we systematically compared the above methods and the integrated method [3] in Appendix A.2.
> 3. We have included a discussion in Section 8 about extending our methods to other graph domains, such as dynamic graphs, heterogeneous graphs, and heterophilous graphs.
> 4. To improve reproducibility, we have released our code at:  https://github.com/liangliang6v6/Multi-GC. We have also expanded the appendix to include more details on our implementation to facilitate reproducibility.

---

### Review · Reviewer_veQG · 2025-02-20

**Summary Of Contributions:**

This paper provides a benchmark study on the graph condensation methods on the multi-label graph datasets. Especially, it discusses and analyses the impact of different initializations and optimizations strategies as well as different graph condensation methods on the multi-label graph.

**Audience:**

Yes

**Claims And Evidence:**

No

**Requested Changes:**

seen above in the weakness.

**Strengths And Weaknesses:**

Strengths:

(1)The paper addresses an important problem in multi-label node classification.

(2)The paper benchmarks existing methods as well as different initialization and optimization techniques, offering valuable insights for graph condensation on multi-label classification.

Weakness:

(1) In section 2.2 related work: The discussion on prior work primarily focuses on studies published before 2019. However, several recent works have explored multi-label graphs. Please include a discussion of these more recent contributions on multi-label node classification to provide a comprehensive and up-to-date perspective.

(2) Notation definition: Ensure precise definitions of all notations used. For example, in Section 4.3, while explaining GCDM, the meaning of C is unclear. Please clarify.

(3) challenges of applying existing methods on the multi-label setting are not clear: in section 6.1, explain more concretely what exactly is the problem of applying the approaches developed on the single-label graphs directly to multi-label graphs. Does same nodes get sampled multiple times in the subgraphs cause a problem? Do class imbalances in sampled subgraphs affect learning? What fundamental issues arise when applying methods from single-label to multi-label settings?

(4) Categorize the methods: in section 6.1, bold the names of methods/settings in the itemized list for clarity. Clearly distinguish between node-only and subgraph-based methods, aligning these categories with later discussions to improve readability and coherence.

(5) In section 6.2, the performance differences across graph condensation methods could be influenced by the connectivity of sampled subgraphs and their homophily levels. Include characteristics of sampled subgraphs under node-only and subgraph-based settings to provide a clearer understanding of these differences. Structure observations by explicitly associating each method with the setting in which it is evaluated (e.g., Method A in Setting S).

---

> ### Author Response · Authors · 2025-03-07
>
> (1) Thank you for the comments. We added more recent works on multi-label graphs in Section 2.2.
>
> (2) We appreciate the careful review. The notation in Section 3 contained a typo $C$ should represent the number of classes, denoted as $K$. We have corrected this in the revised manuscript.
>
> (3) We've clarified in Section 6.1 the challenges of adapting graph condensation methods for multi-label graphs. Unlike single-label graphs, multi-label graphs have nodes with overlapping labels, complicating batch sampling and leading to redundant node inclusion, which biases label associations. Traditional sampling can exacerbate class imbalance, weakening the representation for rare labels. Additionally, single-label loss functions do not capture complex label dependencies necessary for preserving multi-label structures. Addressing these requires adapting these functions to accurately reflect multi-label relationships. We hope these clarifications address your concerns.
>
> (4) We have made the suggested changes to Section 6.1, ensuring that methods and settings are clearly categorized. We have also added precise definitions where necessary to improve clarity.
>
> (5) We are adding the characteristics of subgraphs to show a better comparison in Appendix A.4. Based on subgraph connectivity (Table 6) and homophily (Table 7), we observe that methods preserving higher structural and attribute-based similarity consistently achieve better F1 performance (Table 2). For instance, in datasets like PPI and OGBN-Proteins, the K-Center method maintains higher subgraph connectivity and homophily compared to Random and Herding, resulting in clearly superior predictive accuracy. Conversely, for datasets such as PCG, the Herding method's better homophily and connectivity align with its improved classification performance. These outcomes highlight that maintaining structural coherence and attribute similarity in subgraphs directly supports better multi-label classification results.

---

### Decision · Action_Editor_ttZ4 · 2025-04-13

**Recommendation:** Accept with minor revision

**Comment:**

This paper presents a benchmark study on graph condensation methods for multi-label graph datasets. It specifically analyzes the impact of various initializations, optimization strategies, and graph condensation techniques on multi-label graphs. The proposed benchmark study is insightful and extensive experiments are conducted. The key observations provide valuable contributions to the field of graph condensation. The paper requires revisions and improvements before it can be accepted. The motivation behind the paper should be clarified further, and more related works and state-of-the-art methods should be discussed and incorporated into the benchmark. Additionally, the presentation needs improvement, particularly in reference formatting and clearer notation definitions. Overall, this is a useful contribution, but some revisions are necessary. Therefore, I recommend acceptance with minor revisions.

**Audience:**

The paper will be attract some individuals in the graph machine learning community.

**Claims And Evidence:**

The claims made in the submission are supported by accurate, convincing and clear evidence.